# Emerging and Novel Viruses in Passerine Birds

**DOI:** 10.3390/microorganisms11092355

**Published:** 2023-09-20

**Authors:** Richard A. J. Williams, Christian J. Sánchez-Llatas, Ana Doménech, Ricardo Madrid, Sergio Fandiño, Pablo Cea-Callejo, Esperanza Gomez-Lucia, Laura Benítez

**Affiliations:** 1Department of Genetics, Physiology, and Microbiology, School of Biology, Complutense University of Madrid (UCM), C. de José Antonio Nováis, 12, 28040 Madrid, Spain; chrsan01@ucm.es (C.J.S.-L.); rimadrid@ucm.es (R.M.); pcea@ucm.es (P.C.-C.); lbenitez@ucm.es (L.B.); 2“Animal Viruses” Research Group, Complutense University of Madrid, 28040 Madrid, Spain; domenech@ucm.es (A.D.); sergifan@ucm.es (S.F.); duato@ucm.es (E.G.-L.); 3Deparment of Animal Health, Veterinary Faculty, Complutense University of Madrid, Av. Puerta de Hierro, s/n, 28040 Madrid, Spain

**Keywords:** biodiversity, DNA viruses, emergence, metagenomic studies, passeriformes, RNA viruses, spillover, zoonoses

## Abstract

There is growing interest in emerging viruses that can cause serious or lethal disease in humans and animals. The proliferation of cloacal virome studies, mainly focused on poultry and other domestic birds, reveals a wide variety of viruses, although their pathogenic significance is currently uncertain. Analysis of viruses detected in wild birds is complex and often biased towards waterfowl because of the obvious interest in avian influenza or other zoonotic viruses. Less is known about the viruses present in the order Passeriformes, which comprises approximately 60% of extant bird species. This review aims to compile the most significant contributions on the DNA/RNA viruses affecting passerines, from traditional and metagenomic studies. It highlights that most passerine species have never been sampled. Especially the RNA viruses from *Flaviviridae*, *Orthomyxoviridae* and *Togaviridae* are considered emerging because of increased incidence or avian mortality/morbidity, spread to new geographical areas or hosts and their zoonotic risk. Arguably poxvirus, and perhaps other virus groups, could also be considered “emerging viruses”. However, many of these viruses have only recently been described in passerines using metagenomics and their role in the ecosystem is unknown. Finally, it is noteworthy that only one third of the viruses affecting passerines have been officially recognized.

## 1. Introduction 

An emerging viral disease can be defined as a new occurrence of a disease because of: (a) the evolution or change of an existing virus or its spread to a new geographic area, species or ecological niche; (b) its rapidly increasing incidence, in terms of numbers of infected individuals or geographic range; or (c) a previously unrecognized disease or virus [1,2,3,4,5]. A viral disease of the past (i.e., one previously considered to be controlled) that re-appears with an increased prevalence in an area with susceptible host populations, expands its host range or appears in a new clinical form, is usually termed as re-emergent viral disease [4,6]. Many recent human emerging viral diseases have an animal origin, some with a significant impact on animal or public health, such as SARS-CoV-2, and two viruses that infect passerines: influenza A virus (AIV) and West Nile virus (WNV). The advent of modern, more powerful sequencing and bioinformatics technologies has increased the discovery of novel viruses in animals, with or without causing disease, that may be pathogenic, emergent, or zoonotic. Many of these novel and potentially emerging viruses are found in avian species that are present in virtually every ecosystem. Surprisingly, although passerines are the most abundant and diverse avian species worldwide, little is known about the novel and emerging viruses that they host and their possible role in the emergence of new viral diseases in animals and humans.

Several factors are related to the development of emerging viruses and diseases as they enable infectious agents to evolve into new ecological niches, to reach and adapt to new hosts, and to spread more easily among the new hosts: urbanization and destruction of natural habitats, allowing humans and animals to live in close proximity; international travels and trade; climate change and changing ecosystems; changes in populations of reservoir hosts or intermediate insect vectors [4,5]. All these inter-dependent factors imply that the study of new or emerging viruses must be approached from a multisectoral and multidisciplinary perspective, framed in the One Health approach supported by the WHO, WOAH and FAO [4,5] (Figure 1). The international trade of passerines, mainly ornamental breeds, has been also associated with the introduction of novel viruses in some countries, which pose a risk to native or endemic species if they can jump the species barriers, such as described with avipoxvirus in New Zealand [7]. There is also reasonable concern that more vulnerable individual species (of all taxa, including Passeriformes) may be at risk of extinction from viral pathogens. It is suggested that island endemic species are particularly vulnerable to pathogens, especially introduced pathogens to which they have no prior contact and no innate immunity [7]. Other authors have pointed out that all species with a small geographic range, low population size and low genetic diversity may be highly vulnerable to extinction, not just those island endemic species [8]. However, we are not aware of any evidence that any species has ever become extinct due to a viral pathogen.

One of the most important factors that determines viral emergence is related to the adaptation of a given virus to new host species and/or the concomitant appearance of changes in the environment that offer new opportunities for the virus to thrive [9]. Some emerging viruses have a broad range of hosts. For example, RNA viruses such as West Nile virus (WNV) and Avian Influenza virus (AIV) can infect hundreds of different bird species including passerines. Others, such as herpesviruses or papillomavirus, are usually considered to be specific for one or a few related species [9,10]. Many species may act both as natural viral reservoirs and as amplifying hosts in bird-vector-bird cycles, for example, some togaviruses and flaviviruses. Occasionally these viruses are transmitted to incidental (dead-end) hosts including humans, equids, and other mammals. Birds can also act as a gene source of emerging viruses in cross-species transmission, for example, new influenza viruses may evolve through the reassortment of different gene segments [11].

The potential adaptation of an emerging virus to a new host depends on viral transmission routes (shedding of virus and infection in individuals) and the possibility of reaching this new host is facilitated by close and prolonged contact between individuals (such as breeding facilities). Viral transmission in birds may be horizontal (between individuals) or vertical (from females to offspring, congenitally or through embryonated eggs such as avian leukosis virus). Horizontal transmission is the most frequent and can occur by direct contact between animals (aerosols, fluids, feces, wounds or by predation or scavenging), by indirect contact through fomites or contaminated material (such as water, food, or troughs), and by vectors. Common vectors are blood-feeding arthropods such as mosquitos, midges, and ticks, in which the virus is propagated, and viruses transmitted mainly by this route are collectively called arboviruses (from Arthropod-Borne viruses). Vector-borne transmission is an indirect route of increasing importance in emerging viral diseases, some of which are important zoonoses, such as flaviviruses and togaviruses. It should be noted that viruses can be transmitted by more than one route.

Alternatively, viruses can enter the body via epithelial or the superficial mucosa of respiratory, gastrointestinal, and urogenital tracts. The most common routes of infection are ingestion of contaminated water or food (the fecal-oral route, for instance, picornaviruses) or inhalation of droplets expelled by an infected individual (respiratory route, for instance, herpesviruses and metapneumoviruses) or contaminated surfaces. Respiratory viruses can also be transmitted by contact with eye mucosa. In airborne transmission, viruses can spread over long distances through small respiratory aerosols that can remain suspended and travel in the air, such as influenza virus [12]. Viruses transmitted through ingestion usually are non-enveloped, resistant to low pH and to the acids in the digestive tract, and often produce diarrhea, causing large amounts of virus to be shed into the environment (where they can remain infectious for a long period of time) [9]. Diverse pathogens are probably transmitted between wild birds and domestic birds and poultry when feed and water are contaminated with feces in open aviaries and free-range farms. Describing the cloacal virome is essential for understanding the ecology of viruses circulating in the environment, identifying new virus-host relationships, and defining the risk of virus emergence [13]. However, data on the cloacal virome of wild or domestic passerines are still very scarce and, besides our previous study in French Guiana and Spain [13], there are only a few other studies in China [14,15] and Australia/New Zealand [7,16,17]. Increased investigation of the virome and emerging viruses of passerines is vital for predicting future outbreaks and spillovers that can affect birds and other animals, humans and biodiversity [13,15].

Greater than two-thirds of viral taxa that infect humans are considered to be zoonotic: they are able to infect non-human vertebrates and may circulate in non-human reservoirs [18]. The alternative hosts for most zoonotic viruses are mammals (rodents, ungulates, other primates, carnivores, and bats). Birds are a much less important reservoir for zoonotic disease than mammals. Though less than 20% of zoonotic viruses share avian hosts [18], current data shows that birds are an important potential source for zoonotic viruses.

Passeriformes is the most diverse avian order. These are songbirds and perching birds with well-known species including sparrows, starlings, thrushes, magpies, crows, swallows, and finches. There are around 10,700 bird species, placed in 41 orders and 248 families, of which nearly 6400 species (59.6%) and 140 families (56.5%) are Passeriformes [19], making it by far the most speciose avian family. In addition, many passerine species are also extremely abundant. One recent estimate of the abundance of 92% of bird species determined that there are approximately 50 billion individual birds (albeit with high levels of uncertainty), of which 28 billion (56%) are Passeriformes [20]. This begs the question of whether the diversity of viruses circulating in Passeriformes is approximately equal to their share of avian diversity and abundance, and whether they could pose a significant risk to human and animal health and environmental balance. 

Passerines include wild, urban, rural, and pet birds. Many passerine species are extremely abundant in the wild, but many populations of passerines are synanthropic species that live in close proximity to anthropogenic environments, which boosts the risk of the circulation of viral diseases between humans [21] poultry, and passerines (Figure 1). Pet passerines are usually bred for ornamental use, kept in captivity at home or in different types of aviaries (e.g., breeding facilities) that may have access to outdoors increasing the risk of contact with wild birds [22]. Migratory passerines can spread a wide variety of viruses over long distances, potentially being capable of infecting resident wild passerines (such as house sparrows, *Passer domesticus*), and these latter possibly contaminating pet birds living in open aviaries or poultry in farms [22]. For example, it has been suggested that small passerines could serve as bridge hosts for low-pathogenic avian influenza virus (LPAI) from infectious waterfowl to commercial turkey farms [23]. It is possible that the contact zone between agriculture and wildlife provides an interface where viruses could potentially circulate between wild and domestic birds [23] but probably vice versa as well. Passerines could play a larger role than previously thought in this interaction, a hypothesis that needs to be further studied.

We discuss viruses in 99 species of Passeriformes from 30 families (we also mention in passing a few non-Passeriformes when it is relevant to the virus in question). We emphasize that this review is not intended as an exhaustive list, especially for the best-surveyed, and most common viruses such as certain *Flaviviridae*, *Orthomyxoviridae* and *Poxviridae*. We are aware that discussing so many species may be complicated for readers with little background in ornithology, and we thus provide a list of all species, or groups, of birds (Appendix A), and avian families (Appendix A) cited in this review.

We have found certain passerine families to be more frequently associated with virus detection than others. The most important, in order of frequency, are the following: Family Turdidae (thrushes) which includes large and abundant species, such as the common blackbird (*Turdus merula*) and American robin (*T. migratorius*). Families Estrildidae (waxbills, and allies) and Fringillidae (true finches) include attractive colorful species and are common aviary birds. Most viruses discussed in Estrildidae were detected in captive birds, such as Gouldian finch (*Chloebia gouldiae*) and zebra finch (*Taeniopygia guttata*); the origin of viruses in Fringillidae is more varied, about one-third were detected in the canary (*Serinus canaria* var *domestica*) a very common companion bird, but the majority were detected in species that are expected to be wild, like the common chaffinch (*Fringilla coelebs*). Family Passeridae, the Old World sparrows, including house sparrow and tree sparrow (*Passer montanus*), are mentioned repeatedly. These species are incredibly abundant, and house sparrows are considered to be the most abundant avian species with an estimated population of 1.6 billion individuals [20]. Family Corvidae (crows, jays, and magpies) and Sturnidae (starlings), very large and fairly abundant, passerine families, both seem to be important groups for some viruses. Family Paridae (tits, chickadees, and titmice) are common, brightly colored birds that frequent bird feeders, and are highly visible to bird-lovers.

Therefore, the relevance of passerines in the maintenance of viruses in ecosystems and in the worldwide dissemination of emerging viruses is a topic of growing interest. However, very little is still known about the viruses that affect passerines and their contribution to the emergence of viruses and diseases. One of the reasons that could explain the unknown role of passerines in different ecosystems in the viral circulation could be that asymptomatic individuals in wild populations are usually under-sampled [24] (see flaviviruses and avian influenza virus sections) and data rely on passive surveillance (e.g., reports of unexpected mortality in wild passerines species) [25,26,27]. Moreover, occasionally there is a risk that results of serosurvey are over-interpreted as evidence that passerines may be infected with a specific virus [28], though seropositivity may only evidence a historic immune response to the virus.

This review provides a detailed appraisal of the literature on viruses present in passerines, based on data from serosurveys, molecular surveillance, metagenomic studies and experimental inoculation studies. The data are ordered alphabetically following the unbiassed criteria of the type of viral genome and whether the virion is enveloped or naked. The authors also assess whether the available data demonstrate that the virus group should be considered novel or emerging. Since many of the findings are from metagenomic studies, an effort is made to differentiate those instances in which a link is established between disease and virus from those others that are incidental.

## 2. Families of Viruses Affecting Passeriformes

The most important families of viruses affecting passerine birds are shown in Figure 2. Data on the bird species and families these viruses affect is summarized in Table 1, Appendix A. Species recognized by the International Committee on Taxonomy of Viruses (ICTV) are italicized.

### 2.1. Enveloped Double-Stranded DNA (dsDNA) Viruses

#### 2.1.1. Family *Herpesviridae*


The *Herpesviridae* family is divided into three subfamilies: *Alphaherpesvirinae*, *Betaherpesvirinae*, and *Gammaherpesvirinae* [29] with linear genomes about 125 to 240 kbp. The icosahedral capsid is surrounded by a loose tegument and envelope forming 150–200 nm spherical virions. Avian herpesviruses belong to the subfamily *Alphaherpesvirinae* and can cause well-known diseases in poultry and anatids, such as Marek’s disease, infectious laryngotracheitis, and duck plague [28,30,31], Pacheco’s disease in psittacids [32], and Smadel’s disease in pigeons [33]. 

However, the complete host range of avian herpesviruses in wild birds remains unknown [34], especially in passerines because herpesviruses usually replicate in healthy free-living birds with minimal or no apparent signs of infection. Two field surveys from Slovenia found a prevalence of infection of around 0.8% [35]. Herpesviruses were only detected in five of 46 different passerine species sampled, including the Eurasian blackcap family is divided into three subfamilies: (*Sylvia atricapilla*), common blackbird (*T. merula*), Eurasian blue tit (*Cyanistes caeruleus*), hooded crow (*Corvus cornix*), and European greenfinch (*Chloris chloris*). All partial polymerase DNA sequences are grouped in the subfamily *Alphaherpesvirinae*. Another recent study conducted in free-living birds in New Zealand [7] discovered a herpesvirus from genus *Iltovirus* in song thrush (*Turdus philomelos*) that was most similar to *Psittacid herpesvirus-1* (PsHV-1; 81.5% amino acid identity). No clinical signs were specified in any of these studies [7,34,35].

Generally, clinical manifestations of herpesvirus infection in free-living birds include a wide range of non-specific signs from respiratory to enteric problems [36]. The alphaherpesvirus Passerid herpesvirus-1 was first detected in Central Illinois (USA) in 1998 in a flock of around 700 finches [37], from which 248 Gouldian finches (*Chloebia gouldiae*) and 19 finches of other species died. Cellular abnormalities, including pseudostratified epithelium, hyperplasia, karyomegaly and necrosis were observed in the trachea during necropsy, compatible with herpesvirus infection. The outbreak followed the introduction of 20 Gouldian finches from Canada to the main flock and caused a mortality rate of 38.1% and a morbidity rate of 63.57% among the 20 finch species that suffered mortality. Additionally, 404 finches from 14 additional species developed conjunctivitis and respiratory symptoms. Subsequently, two fatal cases of severe tracheitis and bronchitis were detected in a private aviary in Canada affecting a flock of 20 Gouldian finches [38]. The necropsy revealed intranuclear inclusion bodies in epithelial cells of the upper respiratory tract and the virus was identified by PCR. DNA polymerase sequences of Canadian and US outbreaks showed high identity supporting a link between both.

Although clinical signs of herpesviruses have not been documented in free-living birds, certain herpesviruses have the ability to cross species barriers and establish endemic infections in non-definitive hosts, often resulting in mortality. Nonetheless, cross-species transmission of herpesviruses between birds is rarely observed, and due to their latent nature, it is challenging to establish disease associations [39,40]. *Columbid herpesvirus-1* (CoHV-1), the agent that causes Smadel’s disease in rock doves (*Columba livia*) [41], has been detected in 10 raptor species that displayed clinical signs of infection, and two passerines, a hooded crow (*Corvus cornix*) and a song thrush, without apparent clinical signs [40]. Similarly, interspecies transmission of PsHV-1, the causative agent of Pacheco’s disease, a highly infectious ailment that affects various parrot species and leads to significant mortality rates has been documented in the passerine superb starling (*Lamprotornis superbus*) [26]. 

Despite the limited understanding of emerging herpesviruses in free-living passerine birds [35], they may cause lethal infections in them (Table 1, Figure 2), and may be transmitted to other bird species, potentially resulting in severe and fatal diseases in both intraspecies and interspecies infections. 

#### 2.1.2. Family *Poxviridae*


Poxviruses are large (220–450 nm long), brick-shaped enveloped viruses with genomes of 250–400 kb, which encode 250–300 proteins. Avian poxvirus (APV) infections are among the earliest described avian diseases due to the obvious and sometimes spectacular external lesions. Infection caused by APV can originate discrete cutaneous wart-like growths or lesions, typically detected on the bare areas of birds (for instance, beak, eyes, or legs) although extensive lesions in feathered areas have also been described, for example, in the great tit (*Parus major*) infected with the emerging highly pathogenic Paridae Poxvirus [42,43]. This is a relatively slow-developing disease that is not generally considered to be the direct cause of mortality of affected birds, but the lesions that this disease produces may increase the risk of accident, predation, acquiring secondary infections, and reduce breeding and foraging success [44,45,46]. A second less common form of pox, diphtheritic pox, triggers lesion growth on the mucous membranes, especially of the mouth and respiratory and digestive tracts. This form is considered more lethal [47,48]. Mixed pox infections of both the cutaneous and diphtheritic forms can also occur [49]. Septicaemic infection is a rare form of APV, causing near total mortality, and is mainly reported in canaries [27]. APV is transmitted through direct contact with infected individuals, indirect physical contact (environmental fomites) and via arthropod vectors [27,50]. 

The infection has been detected in at least 374 species from 23 orders of birds worldwide [51] including poultry, songbirds, raptors, parrots, waterbirds, and seabirds among others. Most APV detections (42%) have been from passerine species (mainly finches, sparrows and tits [51] with a highly variable viral prevalence in surveys (excluding epizooties) ranging from 0.8 to 13.5%. However, this is one of the four bird orders (along with Coraciiformes, Caprimulgiformes and Piciformes) where fewer infected species than expected have been detected [43,51]. Additionally, several epizootic events showing very high prevalences in Passeriformes of 50%, in short-toed larks (*Calandrella rufescens*), up to 40% in Hawaiian ‘elepaio (*Chasiempis sandwichensis*) have been documented especially in remote island groups, while two studies on house finch (*Haemorhous mexicanus*) in continental USA revealed viral prevalence of up to 30% [51]. 

APV belongs to the genus *Avipoxvirus*, which contains 12 species based on disease characteristics, host and ecological niche and growth characteristic on cellular culture. The phylogenetic reconstruction identifies three highly divergent major genetic lineages: Clade A (represented by the species *Fowlpox virus*, FWPV), Clade B (*Canarypox virus*, CNPV) and Clade C (*Psittacinepox virus*, PSPV) [52] although some authors have considered two additional clades (D and E corresponding to *Quailpox virus*, QUPV and *Turkeypox virus*, TKPV) [53]. The majority of data from sequence repositories are from Clades A (67%) and B (30%). CNPV (Clade B) was originally described in a canary, but it has been identified in 12 avian orders in addition to Passeriformes. FWPV (Clade A) is mainly found in poultry and to a lesser extent in pigeons, but about 5% of FWPV sequences have been from passerines. FWPV and TKPV are generally limited to galliform birds, but they have been detected in house sparrows that may come into contact with poultry or domestic birds [54]. Submitted APV sequences have been detected in at least 20 passerine families (Table 1) [43]. It appears likely that some APV taxa can infect a restricted range of hosts, while others have an extremely broad host range [43,51]. 

APV is fairly common and has been known as a pathogen of birds for nearly 150 years. It seems to be emerging in some geographic areas, most notably in island groups like Hawaii, Galapagos, or the Canary Islands, and “Paridae Poxvirus” in Europe. Modern sequencing tools are beginning to reveal APV genetics, and suggesting that there are some spillover events and that APV may be expanding the host range.

### 2.2. Non-Enveloped dsDNA Viruses

#### 2.2.1. Family *Adenoviridae*

Adenoviruses (AdV) are icosahedral viruses, about 90 nm in length, with a linear non-segmented DNA genome of 24 to 46 kb. *Adenoviridae* is currently divided into six genera, with three found in avian hosts: *Aviadenovirus*, *Atadenovirus*, and *Siadenovirus.* AdV from these three genera have been isolated from passerines. AdV was first described in passerines in an aviary in the USA housing around 100 Gouldian finches with respiratory problems [55]. Though the process was thought to be caused by circoviruses, viral particles consistent with the size and morphology of adenoviruses were observed by electron microscopy. Adenoviruses from US Gouldian finches and others from Hungary are classified as siadenoviruses [56], with the proposed name Gouldian finch adenovirus-1 (GFAdV-1). This proposal is based on partial DNA polymerase sequences and, in one case, the pre-terminal protein (pTP) gene. GFAdV-1 was also found in the droppings of red-billed fire finch (*Lagonosticta senegala*), red-throated parrot finch (*Erythrura psittacea*) [57], and in zebra finch (*Taeniopygia guttata*) [58]. Siadenoviruses were also detected in a wild great tit (*Parus major*) that died spontaneously in Hungary [59], based on sequence analyses of a 13.6-kb partial genome, and named *Great tit adenovirus-*1 (GTAdV-1). A virtually identical GTAdV-1 sequence was detected in a great tit in Germany [58]. *Siadenovirus* has also been detected in noisy miner (*Manorina melanocephala*), another passerine, in Australia [60]. Since describing GFAdV-1 and GTAdV-1, the number of adenoviruses found in passerines has increased enormously. Passerines seem to be a common source of novel adenovirus sequences [60,61] and over 10% of randomly collected samples from birds are PCR-positive [61]. In addition, results suggest a broad variety of adenoviruses circulating in passerines [58]. A complete adenovirus genome was recovered from the eastern spinebill (*Acanthorhynchus tenuirostris*), and proposed as a new viral species Passerine adenovirus 1 (PaAdV-1) [62]. It is highly divergent from previously known taxa, with *Atadenovirus* genomic organization, the highest sequence similarity (55.6%) to psittacine atadenovirus-3 (PsAdV-3), but 17 predicted novel genes. Novel atadenoviruses have also been obtained from the droppings of long-tailed finch (*Poephila acuticauda*) for which the name Estrildidae adenovirus (EsAdV-1 and -2) has been proposed [57], and in vitelline masked weaver (*Ploceus vitellinus*) and zebra finch (*Taeniopygia guttata*) [58].

Fewer *Aviadenovirus* sequences are known from passerines. Six of 25 adenoviruses detected in a study from Germany were classified as *Aviadenovirus*: three vitelline masked weavers and a great tit from a zoo, one European goldfinch (*Carduelis carduelis*) and one European greenfinch (both wild). Sequences showed 71% identity with marten adenovirus-1 [58]. Adenovirus sequences from Australian Passeriformes have been submitted to GenBank for 11 species of nine families [60]. 

Two observations suggest that avian adenoviruses affect a broad range of hosts. Firstly, identical sequences have been obtained from several passerine species occupying the same habitat, demonstrating that some passerine adenoviruses infect a wide range of passerine species [60]. Secondly, multiple recombination events have been reported between distantly related adenoviruses, which may have occurred in co-infected hosts [63]. Phylogenetic analyses suggest adenovirus-host co-evolution combined with occasional host switches [58].

The clinical significance of adenoviruses in passerines is not yet known. Adenoviruses with varying pathogenicity circulate in commercial poultry [58] and several *Aviadenovirus* may be primary pathogens and economically important. They induce three important diseases in poultry: adenoviral gizzard erosion, hydropericardium-hepatitis syndrome, and inclusion body hepatitis [64]. However, infections in wild birds seem to be generally sub-clinical, lifelong persistent and with continuous or intermittent shedding [57,60] or acting as opportunistic pathogens which may also have the potential to trigger secondary disorders [58]. 

#### 2.2.2. Family *Papillomaviridae*


Papillomaviruses are circular double-stranded DNA, with a genome of about 8 kb, a size of about 60 nm and replicate in the stratified and differentiated epithelium of the skin and mucosa. The first reports of papillomatosis in common chaffinch date back to the 1960s, although the viral origin was not recognized [65,66]. The lesions produced are scaly papillomas, described as cauliflower-shaped neoplasms, which grow on the tarsi and digits, deforming the normal appearance of the leg with common keratinization of the epithelium. Chaffinch tumors are reported to weigh up to 5% of their hosts’ body weight and were found in 1.3% of 25,000 individuals [67]. Tumors result in digit [66] and limb loss [68]. Lesions similar to papillomas containing papilloma-like viral particles have been described in the toes, legs, or the beak commissure of several passerine species, including *Chloris chloris*, *Fringilla montifringilla* and *Serinus canaria* [67,69,70,71,72,73]. 

To date, avian papillomaviruses (PV) have been recovered from 12 avian species, corresponding to at least 13 unique PV types [10]. Only two of these PV types were detected from passerines, FcPV1 (*Fringilla coelebs papillomavirus 1*) [74] and ScPV1 (*Serinus canaria papillomavirus 1*), the first oral avian papillomavirus [68]. Both share 60–70% similarity and are included in the genus *Etapapillomavirus*. The taxonomic classification is based on the nucleotide sequence of the L1 ORF which encodes the major capsid protein. The other APVs belong to four different genera and share less than 60% L1 ORF identity. PV-like tumors have been observed in several passerine species, although virus sequences have not been recovered. Future studies should explore if these represent new PV taxa in passerines, or if the PV host range is broader than currently appreciated.

### 2.3. Single-Stranded (ss)DNA Viruses

#### 2.3.1. Families *Circoviridae*, *Anelloviridae* and *Genomoviridae*

The families *Circoviridae* and *Anelloviridae* are comprised of viruses with circular, covalently-closed, single-stranded naked DNA genomes sized from 15 to 30 nm, with small genomes of 1.8–3.8 kb. They are commonly associated with immunodeficiency-related diseases that affect different mammals as well as birds, freshwater fish, and invertebrates. The family *Circoviridae* has been classified into two genera, *Circovirus* and *Cyclovirus* with about 50 species in each. Among the most common infections caused by circovirus in poultry and wild birds are infectious psittacine beak and feather disease (BFD) and circovirus disease of pigeons, while chicken anemia virus (CAV) is the best-known member of the anelloviruses (genus *Gyrovirus*) that can cause an acute infection of chickens worldwide.

Beak and feather disease virus (BFDV) can cause acute or subclinical disease in captive and wild psittacine species. Infection is an important cause of morbidity and mortality, characterized by feather and beak deformities. Other nonspecific clinical symptoms include diarrhea, anorexia, depression, and immunosuppression [75]. BFDV is a host-generalist widely known in psittacids with a host range of over 370 avian species, including non-psittacid species [76]. It has been detected in Passeriformes from four Australian magpies (*Gymnorhina tibicen*) and one Australian raven (*Corvus coronoides*) from an Australian Wildlife Health Centre [76], and also from a captive flock of Gouldian finches with feather lesions [77], two zebra finches and a Java sparrow (*Lonchura oryzivora*) from a live-bird market in Bangladesh [78]. Plausible BFDV sources for Passeriformes include ingestion of BFDV-infected birds, ingestion of BFDV-contaminated insect vectors, or exposure to a BFDV-contaminated environment [76].

Five recently recognized *Circovirus* species by ICTV have been identified in Passeriformes (*Circovirus Canary*; *Circovirus Finch*; *Circovirus Raven*; *Circovirus Starling*; *Circovirus Zebra finch*) in birds with feather disease or immunosuppression [25,79,80,81,82]. *Canary circovirus* (CaCV) has been traditionally linked to a condition known as “black spot” that affects neonatal and young canaries, characterized by abdominal enlargement, gallbladder congestion, yellowish fluid in the air sacs and growth retardation [83]. Circovirus infections have been also documented in adult canaries that died following a short illness characterized by dullness, anorexia, lethargy, and feather disorder [25]. However, little is known about the epidemiology of the disease.

Although only a small number of viruses have been described in this group of birds so far, the number of new avian gyroviruses, circoviruses, genomoviruses and diverse single-stranded DNA viruses known as Circular Rep-encoding single-stranded (CRESS) DNA viruses [84,85,86,87] is increasing thanks to high throughput sequencing, alone or in combination with rolling amplification techniques. However, an inherent problem in the identification of viruses in metagenomic analyses is the scarcity of data on the clinical signs of disease in avian hosts or their relationship with any pathology. A very significant example is the identification of 108 viruses in fecal samples from 38 endemic New Zealand robins (*Petroica australis*) [17]. The viruses belong to the families *Circoviridae*, *Genomoviridae* and *Microviridae*, although numerous unclassified CRESS DNA viruses have also been found. Likewise, 27 viruses belonging to the *Genomoviridae* family have been characterized from 21 species of passerines in China [88]. Although none of these viruses have been associated with any disease in passeriformes, Gy11 gyrovirus appears to be circulating in at least three species of passerines (*Myrmoderus ferruginea*, *Philydor erythrocercum and P. albifrons*) resident in a remote rainforest in French Guiana with practically no human influence, although with lower prevalence than found in poultry [89]. The virus has been detected in cloacal and oral samples, as well as in blood. The bird likely infected by Gy11, a ferruginous-backed antbird (*Myrmoderus ferrugineus*) was also found to be positive for astrovirus, which may suggest a compromised immune system and a possible co-infection. The presence of this new gyrovirus in oral and cloacal samples from different bird species strengthens the hypothesis that the fecal-oral route is likely the main route of transmission for gyroviruses.

#### 2.3.2. Family *Parvoviridae*


The family comprises non-enveloped viruses, around 18–26 nm in diameter, with linear ssDNA commonly associated with gastrointestinal diseases [90]. These viruses have a genome size ranging from 4 to 6 kb and their genomic organization consists of two main open reading frames [91]. Parvoviruses exhibit a broad host range [92] and can cause significant diseases in poultry or waterfowl, mainly by Goose parvovirus (GPV) [93]. Despite their veterinary importance [94], there is limited information available on parvoviruses in free-living birds. The study conducted by De Souza et al. [95] in Brazil is the sole report thus far, which identified the presence of a parvovirus in a wild grey pileated finch (*Coryphospingus pileatus*) using metagenomics.

### 2.4. Enveloped Negative-Sense Single-Stranded RNA (ssRNA) Viruses 

#### 2.4.1. Family *Orthomyxoviridae*

The family includes viruses with a genome of about 14 kb distributed in eight segments that encode 10–14 genes. Virions are 80 to 120 nm in diameter and each segment constitutes a nucleocapsid. It includes four major influenza viruses (types A to D). Influenza A virus (IAV) infects a broad range of birds and mammals, including humans, and is especially associated with two aquatic bird orders, Anseriformes (ducks, geese, and swans) and Charadriiformes (gulls, terns, and waders/shorebirds) [96,97]. IAV produces a respiratory disease with airborne transmission in humans. In birds, replication occurs in the lung and the gut; ducks are known to excrete the virus for long time periods and are usually infected via contaminated water [96]. Two large surface glycoproteins, hemagglutinin (H), which is responsible for cell-binding and fusion, and neuraminidase (N), which promotes the release of viral progeny, are highly divergent. Bird IAV may express 16 H types (H1–H16), and nine N types (N1–N9) [98]. Bird strains of IAV are known as Avian Influenza Virus (AIV). Due to the reassortment of the segmented influenza genome, which can cause an antigenic shift, there are 144 possible H-N combinations, at least, in theory, and each combination is considered a different subtype. AIV pathogenicity is defined either by the H sequence or by the intravenous pathogenicity index in six-week-old domestic fowl [5], which probably does not cause disease of equal severity in all other bird or mammal species. Some subtypes are considered low pathogenic avian influenza (LPAI) or low pathogenicity (LP), while others are high pathogenic avian influenza (HPAI) or high pathogenicity (HP). Only H5 and H7 combinations are considered HP [97], though not all their combinations are HP. 

As there are potentially 144 AIV subtypes potentially infecting the 6400 species of Passeriformes [19], the discussion in this section is deliberately abridged and concentrates on general themes. The most important aspect of avian influenza for human health is the potential for a recombination, reassortment or mutation that enables the spread of a significant pathogen in the human population. But AI disease is also extremely important economically, for the poultry industry [99]. Another major reason for studying AI in Passeriformes is to focus on the birds that can provide a transmission link from the waterbird reservoir to poultry and back [100], and potentially from areas that are seriously affected by HPAI to areas that are not, through long-distance migration. It is important to understand the transmission risk posed by synanthropic or peridomestic birds, including Passeriformes, such as house sparrows or Eurasian starlings. The importance of Passeriformes to AI transmission cycles is controversial. Some studies find that passerines have a low or negligible contribution to AI transmission [101,102,103,104]. Other studies have found more varied results, including high prevalence or seroprevalence in certain Passeriformes species [105,106,107,108]. A recent study from Germany, with data from 2006 to 2021, including the 2020 to 2021 highly pathogenic AIV outbreak found only 11/972 (1.1%) passerines (all Corvidae) to be positive, compared to 3351/4583 (73%) of all birds tested [109].

One comprehensive review [110] analyzed the role of seven synanthropic passerine families, finding that Passeridae (*Passer domesticus* and *P. montanus*) are susceptible to HPAI, and probably contribute to transmission cycles through moderate virus shedding, and contact transmission, but that their role is not so clear as regards LPAI. Other studies showed that the family Turdidae had either high AI prevalence/seroprevalence [106] or high susceptibility to infection, that Turdidae can experience high viral titers in lung tissue [111] and can shed high levels of the virus [112] in infection studies. This suggested species such as American robins (*Turdus migratorius*) and common blackbirds are probably important to synanthropic transmission cycles. Corvidae, Hirundidae and Icteridae were reported to probably be important to these cycles, though more studies are required to confirm this. On the other hand, they determined that European starlings (*Sturnus vulgaris*) are probably not important as they are regularly infected by IAV, but viral shedding and contact transmission is infrequent in this species, though this may vary by viral strain. The authors found no evidence that Fringillidae are important for the AI transmission cycle [113]. Another study determined that red-billed quelea (*Quelea quelea*), a species that numbers at least 1 billion individuals, is an ideal transmission bridge from waterbird AIV reservoirs to poultry [114]. A red-billed quelea infection study confirmed they are susceptible to HPAI-H5N infection, shed moderate levels of virus for several days, and show lower mortality than the other species in the study (*Silvia atricapilla*), suggesting that they may be important to the transmission cycle [115].

In the European Union and UK 789 HPAI virus detections were reported between 11 June and 9 September 2022, mainly in colonial seabirds (28 species, 541 detections), but also smaller numbers of waterfowl (11 species, 109 detections), raptors (11 species, 38 detections) and other species (15 species, 101 detections), of which at least 51 (6.5%) were from Passeriformes [116]. In North America, the USDA wild and captive bird surveillance detected HPAI 7098 times in 18 avian orders in 2022–2023 (compared to 836 outbreaks in commercial and backyard poultry flocks affecting 58.8 million birds) [117], with 60.8% in Anseriformes, 22.8% in Accipitriformes, and a mere 2.3% in Passeriformes. There is a very clear weight of evidence that waterbirds dominate HPAI transmission cycles, but Passeriformes may also be involved. Evidence shows that some Passeriformes species are susceptible to experimental infection with HP-H5N1, and experience oral and cloacal virus shedding, which may be prolonged and with quite high viral titers [111,115,118]. Furthermore, there may be under-reporting bias for HPAI in Order Passeriformes: waterbirds are the known reservoir for AIV and thus the target for AI surveillance; in general, waterbirds are much larger than Passeriformes, and dead waterbirds are more easily detected, especially mass die-offs of colonial species. 

Present evidence thus suggests that some Passeriformes species, including highly abundant species (e.g., house sparrow, tree sparrow and red-billed quelea) may be HPAI reservoir hosts, but that further investigation is required to evaluate this role.

#### 2.4.2. Family *Paramyxoviridae*

Paramyxoviruses are large spherical viruses (about 150 nm in diameter) with a helical nucleocapsid, which contains the ssRNA genome of around 15 kb in size. The family is divided into four subfamilies. The subfamily *Avulavirinae* comprises 22 accepted species that infect birds, denominated avian paramyxovirus 1–22 (APMV-1 to -22). 

One of the most important paramyxoviruses in the poultry industry is the Newcastle disease virus (NDV) which causes a highly pathogenic viral disease in birds. The ICTV was officially renamed *Avian avulavirus 1* (APMV-1) in 2016, although here we will use the classical name. NDV pathogenicity varies greatly by strain, from non-pathogenic to fatal. The virus can cause lesions in the central nervous system and the urinary, alimentary, and respiratory tracts. Virulent NDV strains infect at least 250 species of wild birds and poultry and outbreaks are reported globally [119,120].

Wild birds, especially Anseriformes, are the main carriers of non-pathogenic NDV variants [121] but highly pathogenic strains have been isolated from cormorants in North America and Central Asia, some linked to mass avian mortality events [122]. However, the effect on passerines is poorly understood. A survey of four species of passerines (*N* = 543) from Helgoland Island (North Sea) during the autumn and spring migration found very low NDV prevalences, with only six birds (1.1%) carrying strains considered non-pathogenic [113]. A study from Turkey [123] found a 0% NDV prevalence in 282 wild passerines of 27 species, while another from Brazil detected no NDV in 64 wild and 30 captive passerines [124]. A study from the USA detected no virus, but antibodies in three of 154 passerines (1.9%): two American crows (*Corvus brachyrhynchos*) and one European starling. The prevalence was much higher in Suliformes (cormorants: 44.9%), and four other avian orders [125]. A study from Kazakhstan recovered NDV sequences from 7/295 passerines (2.4%), though prevalences were much higher in Galliformes (54.5%), Columbiformes (50%) and Suliformes (23.2%) [126]. These studies suggest that passerines have a minor role in NDV transmission. In contrast, a recent survey of more than 1600 birds, including 165 passerines and pigeons (of 15 and three species, respectively) showed a 10.9% prevalence of the NDV sub-genotype V.3 [127]. Unfortunately, this study did not specify the relative contribution of each group to prevalence. 

APMV-2 (previously Yucaipa-like virus) appears more common than NDV in wild birds, particularly passerines and parrots. It appears that psittacids suffer high mortality, while passerines experience mild, self-limiting disease [128]. APMV-2 is found in passerines from a broad geographical area, including Africa, Central America, and Europe. Prevalence was low in Africa: 3.7% of captive passerines (all wild-caught individuals from family Ploceidae) in Senegal [129], and 5.2% of yellow wagtail (*Motacilla flava*) in Kenya [130]. Prevalence was much higher in Europe: up to 31% of free-living birds in Germany [131], and 69% of house sparrows in Spain [132]. APMV-2 has been sequenced from a rufous-collared sparrow (*Zonotrichia capensis*) and southern house wren (*Troglodytes musculus*) in Costa Rica [133], and from a Eurasian robin (*Erithacus rubecula*) in Slovenia [134].

APMV-3 has been isolated from captive finches (*Ortygospiza atricollis* and *Poephila cincta*), which showed clinical signs of disease including apathy, diarrhea, conjunctivitis, and dysphagia [135]. Rare cases of severe illness are reported in finches [136]. Inoculation of five breeding exotic finches from various species with isolate 840/85 serotype 3 caused mortality in two individuals (*Poephila cincta* and *Amandava amandava*), while three remained clinically healthy throughout the six-week study [135]. 

APMV-4 detection in wild and domestic birds is increasing worldwide, particularly in waterfowl [137]. It was reported once in a wild Eurasian starling from Ukraine [138].

To summarize, NDV and APMV 2, 3 and 4 have been detected in passerines, causing disease in some cases. However, APMV do not appear to be very common in passerines, and at present there is no strong evidence of APMV emergence in the group.

#### 2.4.3. Family *Pneumoviridae*


The family, with morphological and genetic characteristics similar to the paramyxoviruses, has garnered attention in recent years due to its emergence as a clinically relevant viral group associated with respiratory tract infections. It is classified into two genera: *Metapneumovirus* and *Orthopneumovirus*. Metapneumoviruses have been identified in birds and humans. *Avian metapneumovirus* (aMPV) is transmitted directly and causes upper respiratory tract diseases and reproductive disorders mainly in poultry, and its epidemiology, dispersal patterns, and genetic evolution are not widely known [139]. The rapid and wide dispersion of aMPV has been associated with migratory bird routes in Europe and North America, and also with international trade [140].

The potential role of wild birds as reservoir hosts in aMPV epidemiology is poorly understood There is very little evidence that pneumoviruses infect or are emerging in passerines. aMPV has been identified in house sparrows and starlings in the USA (Minnesota), coinciding with an outbreak in a domestic turkey farm [141]. The gene encoding the matrix (M) protein was genetically similar (96% predicted amino acid identity) to turkey isolates, suggesting that they may be involved in the spread of the virus [142]. Experimental infections with this virus suggest pigeons and house sparrows can spread the virus in farm conditions to chickens, although they are not natural hosts for aMPV, given that they failed to seroconvert. While chickens showed minimal tracheal rales, pigeons and sparrows remained asymptomatic [143]. 

#### 2.4.4. Family *Bornaviridae*

The family consists of a genetically diverse group of spherical viruses (from 70 to 130 nm in diameter) linear ssRNA genomes of about 9 kb which have been detected in mammals, birds, reptiles, and fish. Recent evidence shows some *Orthobornavirus* taxa cause neurologic and intestinal disorders in birds, mostly psittacines suffering from proventricular dilatation disease (PPD) [144,145]. However, molecular evidence of viral infection has been reported in passerines and aquatic birds. The first bornavirus in Passeriformes was detected in the Eurasian jackdaw (*Corvus monedula*) in Sweden [146]. *Orthobornavirus serini* has been found to be widely distributed in captive populations of another passerine, domestic canary, in Germany [147]. Unlike psittacine bornaviruses, the three genotypes identified (CnBV-1 to CnBV-3) appear to have a rather narrow host range and they have been detected exclusively in canaries, thus far. *Orthobornavirus estrildidae* (EsBV-1) has been recovered from black-rumped waxbill (*Estrilda troglodytes*) and a yellow-winged pytilia (*Pytilia hypogrammica*) in Germany [148], and white-rumped munia (*Lonchura striata*) in Japan [149].

The pathogenic potential of passerine bornavirus is unclear. Experimental infection of common canary does not result in a clear clinical syndrome, whereas natural infection results in a PPD-like disease [147,148].

### 2.5. Enveloped Positive-Sense ssRNA Viruses

#### 2.5.1. Family *Flaviviridae*

Flaviviruses are spherical viruses with a diameter of around 50 nm and a genome of 9–11 kb. The envelope glycoproteins are tightly adhered to the lipid bilayer, and the particle has an icosahedral morphology. The family comprises 97 viral species grouped in four genera transmitted by arthropods, especially mosquitoes and ticks. Most flavivirus are primarily hosted by mammals or birds. Infections range from asymptomatic to severe or fatal hemorrhagic fever or neurological disease. 

The Japanese Encephalitis virus (JEV) group is a well-supported sub-clade within the mosquito-borne virus branch of the genus *Orthoflavivirus*. It is a group of 8–10 taxa, most notably JEV and West Nile virus (WNV), which are closely associated with wild birds [150,151]. They are most frequently vectored by Culicine mosquitoes, in an enzootic cycle in which bird species are amplifying hosts, and other species, such as humans, are considered to be dead-end hosts [152], i.e., they can become infected, but the viremia is insufficient to sustain the transmission cycle. WNV and Usutu virus (USUV), are considered both to be emerging infectious viruses and to be particularly associated with passerine birds. 

WNV is an excellent example of an emerging infectious disease: it is expanding beyond its former geographical and host ranges, and novel lineages are surfacing. WNV was first discovered in West Nile Province, Uganda, in 1937, in the blood of a febrile woman who survived infection [153]. It was not considered very severe until the 1990s, with several major outbreaks in birds, horses, and humans in Mediterranean countries, punctuated by a human outbreak in Romania in 1996 marked by 393 hospitalized cases and 17 fatalities [154]. Prior to 1999, WNV circulated in Africa, southern Europe, western Asia, and Australasia. In 1999 it surprisingly emerged in New York City, causing neuroinvasive disease in humans, captive and wild birds, and then spread widely through the Western Hemisphere: from New York to Argentina and Alaska. The WNV genotype that emerged in New York was virtually identical to a lineage 1 strain identified in Israel in 1998. Once present in the USA, WNV dispersal was facilitated by bird movements, including transcontinental migration [155]. Since 2004, WNV has re-emerged in Europe, with cases spreading out from Hungary, and reported from at least 23 European countries [156] with particularly large outbreaks in Greece, Hungary, Italy, and Romania. Most human cases in Europe from 2004–2017 were WNV lineage 2 [157]. WNV is principally a vector-borne virus, and the main vectors are culicine mosquitoes, particularly from the genus *Culex*. Additional transmission routes, outlined below, have also been observed.

WNV is particularly associated with passerine birds, in part because Corvids, like the American crow (*Corvus brachyrhynchos*) and blue jay (*Cyanocitta cristata*), suffer high mortality rates when infected. Dead WNV-infected crows were commonly reported when the virus emerged in North America [158]. WNV has also been detected in wild passerines in a number of studies from Europe [159,160,161], where both lineages 1 and 2 are circulating. However, in Europe, WNV, especially lineage 2, seems much more associated with Accipitriformes (birds of prey). This may be sampling bias, as large birds of prey housed in Avian Rehabilitation Centres are commonly sampled. However, several large-scale studies of passerine birds have recovered WN antibodies or viral sequences from only a small number of passerines. By contrast, one study from Austria and Hungary found 35/57 (61%) Goshawk (*Accipiter gentilis*) WNV Lineage 2 positive by RT-PCR, compared to seven individuals of six passerine species (*N* = 16; 44%). Goshawk, a relatively rare bird, are predators of passerine birds [162], and may be infected when feeding on passerine prey.

An experimental infection study with the New York 1999 strain of WNV [152], which targeted 25 avian species of 10 avian orders found that all passerine bird species (*N* = 104 individuals of 10 species) were competent reservoir hosts; indeed, the five most competent reservoirs were all passerine species. Seven non-passerine species were not competent reservoir hosts, and eight showed responses varying from weakly to highly competent (*N* = 74 individuals of 15 species). All species were exposed to infected mosquito bites; additional individuals from some species were also exposed to oral or contact infection. Experimental infection caused mortality in seven of 10 passerines, but only in one of 15 non-passerines. Three passerine species (*Corvus brachyrhynchos*, *Pica hudsonia* and *Haemorhous mexicanus***)** suffered 100% mortality. All passerines showed high levels of viremia, generally higher than in non-passerines (Figure 3). All individuals that survived infection developed neutralizing antibodies. Oral transmission was shown in four passerine species. Contact transmission was shown in eight individuals of three *Corvidae* species, but only one non-passerine, when mosquito-infected subjects were placed in a cage with non-infected cage mates. This part of the study involved WNV-free individuals of nine passerine (*N* = 36) and nine non-passerine species (*N* = 24). Viral shedding was demonstrated in nine of 10 passerine species. Viral shedding is likely to contribute to oral transmission of WNV as it frequently circulates in high quantities in both cloacal and oral material [152]. It is likely that WNV is transmitted via contact in natural settings, especially in the most susceptible species, which display the highest viremia and shed the highest levels of virus. A review of experimental WNV infection studies in 77 wild bird species (including 35 from Passeriformes), from 29 families and 12 orders since 1955 [163] concurred that Passeriformes (and Charadriiformes) are generally highly competent WNV hosts.

There is considerable experimental evidence that the pathogenicity of different WNV strains varies in the same bird species. For instance, another experimental infection study of American crows using three lineages from Kenya, Australia and North America showed that the American genotype resulted in a higher serum viremia and more deaths than others. Similar results have been shown in infection studies in house sparrows [152,164,165,166,167,168] and other species [163].

WNV has been reported in more than 300 species of birds, including wild, domestic, captive, and invasive species [169]). The list now stretches to a minimum of 392 bird species, from 27 orders, in which WNV has been detected, by serology or by PCR, as shown in Appendix A, though the real number is likely much higher. Passeriformes (156 species and 39.6% on our list, from 27 families) were the most frequent host species order of WNV. However, since Passeriformes account for 60% of bird species [19], the number of known passerine hosts of WNV is lower than anticipated. It is unclear if this is due to sampling bias, or biological characteristics. Passeriformes are generally small, and often cryptic, and they may be under-sampled. On the other hand, large, charismatic birds, like hawks and ducks, may be over-represented in sampling, especially where surveillance focuses on bird rehabilitation centres, aviaries, and farms. 

Usutu virus (USUV) circulates in wild bird amplifying hosts, and is vectored to new hosts by blood-sucking arthropods, especially Culicine mosquitoes. It was first discovered in Ndumu, South Africa in 1959, in *Culex univitattus*, close to the Usutu River [170]. It was thought to circulate only in Africa, and neighbouring Israel, until it was recovered in Austria in 2001, in association with mass avian mortality, especially in the passerine common blackbird [171]. Since then, USUV has been detected in at least 16 European countries, in mosquitoes, birds and humans [172]. 

Phylogenetic analyses of USUV genomes reveal six distinct lineages denominated Africa 1–3 and Europe 1–3. The most recent common ancestor (MRCA) for the African lineages is estimated to have arisen before the early 16th Century, possibly in South Africa, while the MRCA of the European lineages may have arisen in Western Europe since the 1950s. European lineages probably evolved in Europe [173]. 

Despite the acknowledged association with birds, USUV has been rarely detected in African birds; to the best of our knowledge, only six times [174,175], and just twice from Passeriformes (*Eurillas virens* and *Turdus libonyana*). By contrast, many European studies have demonstrated USUV or antibodies in birds, especially in Passeriformes. Furthermore, in many studies, the vast majority of USUV-positive hosts are common blackbirds: 28 of 30 positive dead birds in Austria in 2001 [171]. Another study also revealed the presence of USUV in 102 of 107 positive dead common blackbirds, compared to 0 of 40% in non-Passeriformes [172,176]. One study on 15 common bird species in Germany, (all but one passerines) determined that common blackbird populations declined by 15.7% more inside USUV-suitable areas than in USUV-unsuitable areas, but found no significant effect for the other 14 species investigated [177]. Occasionally USUV prevalence is very high in other bird species, e.g., house sparrows [178]. The above-mentioned avian virus study from Germany, including data from 2006 to 2021, found USUV in 917/972 (94.3%) passerines to be USUV positive, compared to 1042/4583 (22.7%) of all birds tested [109]. The vast majority (82.2%) of positive birds were common blackbirds.

USUV has been detected in at least 100 bird species (Appendix A), 46 species (46%) and 17 families from Passeriformes, somewhat lower than the proportion of bird species within the order. The number of USUV-positive Anseriformes and Strigiformes species (10.3% and 16.1% respectively) is considerably greater than the proportion each order makes of all bird species (1.7% and 2.2%). There is thus considerable evidence that USUV infects numerous Passeriformes species, and causes pathogenicity and mortality in many, especially common blackbirds. However, the virus has an extremely broad host range, and clearly also infects many non-Passeriformes species.

JEV is an important zoonotic virus, which can cause neuroinvasive disease in humans, infecting about 68,000 people annually with a fatality rate of about 30% [179]. It is considered to be an emerging virus due to the current range expansion, notably in Australia [180]. Wild birds, especially herons (Order Suliformes) and wild pigs are considered to be the main amplifying hosts [181,182]. Occasional JE surveillance in passerines suggests they may also be involved, for instance, 24.1% of tree sparrows (*Passer* montanus) tested in Japan were seropositive for JEV antibodies [183]. A JEV experimental infection study that challenged birds of eight orders, including several passerine species, and two heron species, determined that several passerine species and one gull had the highest average peak viremia titers when challenged with JEV genotype I [182]. The role of passerines in JEV is poorly understood, but it is likely understated.

Murray Valley encephalitis virus (MVEV), from Australia and New Guinea, and Cacipacore virus (CPCV), from Brazil, are also associated with birds. MVEV is typically associated with water birds, especially herons [184], while CPCV was first isolated from a passerine, black-headed antbird (*Percnostola rufifrons*), and Passeriformes are considered the main amplifying hosts [185].

Saint Louis Encephalitis Virus (SLEV) is a reemerging arbovirus in the family *Flaviviridae* [186], which can cause neuroinvasive disease and mortality in humans [187]. SLEV is widespread in the Americas and more than 50 human outbreaks, with a mortality rate of 5–20% have occurred in the USA and Canada, since its first identification in 1933 [188]. 

SLEV cycles between *Culex* mosquito vectors and passeriform and columbiform birds as amplifying hosts [189]. Several mammal species might be involved in the Neotropics [190]. The passerine role as an SLEV amplification host has been demonstrated by several studies [191,192,193]. One large-scale serosurvey found 13,044/24,000 peridomestic birds (54.4%), from five passerine species, seropositive for SLEV [194]. A study from Argentina showed the possible relationship between the 2005 SLEV human epidemic in Cordoba and the presence of SLEV in wild birds, as seroprevalence rose from 0.7% to 7.7% immediately prior to the human outbreak [195]. Almost 6% (25/425) of individuals of 11 passerine species were seropositive, compared to 17/118 individuals (14.4%) of five non-passerine species. These results suggest that passerines are important to SLEV transmission, but possibly less so than other bird groups. They also suggest that wild bird surveillance may provide an important early warning about the SLEV presence and human epidemic risk [196].

A number of other *Flavivirus* taxa are also associated with birds, and in some cases with Passeriformes [197,198]. These viruses belong to five additional Flavivirus groups. Ilheus (ILHV) and Rocio (ROCV), both circulating in Brazil and closely related to SLEV are both best known from passerines. Several Flaviviruses from the Ntaya virus group, including Bagaza, Israel turkey meningoencephalomyelitis, Ntaya, and Tembusu, but these are mainly found in non-passerines, especially Galliformes. Three viruses from the yellow-fever group, Bouboui, Uganda S (USV), and Saboya, have been detected from birds—USV from the passerine *Saxicola rubetra*. Five viruses from the tick-borne encephalitis group have been detected in birds, including Tyuleniy and Meaban from seabirds, and Kyasanur Forest disease, Louping ill, Tick-borne encephalitis, all mainly from non-passerines. Finally, three viruses from the unknown vector group: Phnom Penh bat, Cowbone Ridge, Sal Vieja (SVV)—the last known from 51 bird species [197].

In general, the host taxonomy of the *Flavivirus* taxa is poorly delineated. The best-known taxa, e.g., WNV, USUV, and JEV, adapt successfully to scores of host species and are prone to rapid evolution. Many *Flavivirus* taxa are highly pathogenic, causing neuroinvasive and/or hemorrhagic fever in multiple hosts. This group is highly likely to yield future emerging diseases, many of which probably currently circulate in passerine hosts or will do in the future. Many are known zoonoses.

#### 2.5.2. Family *Togaviridae*

Togavirus are small (around 65–70 nm in diameter), enveloped RNA viruses with icosahedral morphology. The positive-sense ssRNA genome is 9.7 to 11.8 kb long. The sole genus, *Alphavirus*, contains 32 viral species organized into seven species complexes. Alphaviruses are mosquito-borne viruses with high zoonotic risk, capable of producing high mortality and morbidity rates in humans and animals. Alphavirus infections can cause systemic or neuroinvasive disease, characterized by arthralgias/myalgias and meningoencephalitis, respectively. Classically they are described as Old and New World viruses, although some have a worldwide distribution. Vertebrate hosts include humans, frequently as dead-end hosts, other mammals, birds, amphibians, reptiles, and fish. At least eight alphaviruses have been described in passerines. 

Eastern equine encephalitis virus (EEEV) is an enzootic alphavirus [199]. Numerous EEEV epidemics have caused significant mortality and morbidity in humans and animals across the Americas. EEEV incidence in humans and geographical distribution is increasing in the USA [200]. EEEV is transmitted in an epizootic cycle principally involving mosquito *Culiseta melanura* and avian hosts. Other mosquito species may serve as bridge vectors to infect humans, horses [201], and other animals [202,203,204,205]. Although EEEV has a broad host range, passerines are the main amplifying hosts. Studies from New England and Alabama show *C. melanura* preference for feeding on passerines: evidence of EEEV infection has been shown from at least 44 passerine species [206,207]. Some passerines, especially wood thrushes (*Hylocichla mustelina*) and American robins may be EEEV superspreaders [208]. EEEV can cause neurological disease and mortality in passerines, though disease in amplifying hosts is considered uncommon [209]. 

Highlands J virus (HJV) has been detected in North and South America. It can infect humans and horses and is mainly transmitted by *C. melanura* to wild passerines that inhabit fresh-water swamps, apparently causing relatively mild effects [210].

Sindbis virus (SINV) can cause zoonotic disease in humans [211]. Since its initial detection in Sindbis district, Egypt, in 1952, numerous cases and outbreaks have been reported from Africa, Australia, and Eurasia, with high morbidity rates [212]. The SINV enzootic cycle involves *Culex* spp. mosquitoes. Passerines are the most important SINV hosts, though high seroprevalences have also been reported from Anseriformes and Galliformes [213,214]. The highest seroprevalences have been reported from the passerine genus *Turdus* [215,216]. 

Ross River virus (RRV) and Mayaro virus (MAYV) are two zoonotic alphaviruses in the Semliki Forest serocomplex. RRV is endemic in Australia and the south Pacific region [217] and is considered the most widespread arboviral disease in Australia [218]. It was first isolated in 1965 from three passerine species (*Grallina cyanoleuca, Microeca fascinans* and *Poephila personata*) in Australia [219] and has been detected again recently from *Microeca fascinans* [220]. Scarce serological surveys of domestic and wild birds in Australia reveal low infection rates [221]. However, mammals appear more important to transmission cycles than birds [217]. MAYV was first isolated in 1954 in Trinidad [222] and has caused outbreaks in several countries of the Caribbean and South America [223]. The virus is maintained in a sylvatic transmission cycle near forested areas involving mosquito vectors and non-human reservoirs [224]. MAYV antibodies have been detected by hemagglutination inhibition in 21 wild bird species from the orders Charadriiformes and Passeriformes (seven families) in the USA, Brazil, and Colombia [225,226,227]. However, the virus has only been isolated from a passerine species, orchard oriole (*Icterus spurius*), in Louisiana [228]. There are no recent studies on potential non-human reservoirs, although the need for increased surveillance has been emphasized [229].

Buggy Creek virus (BCRV) (also called Fort Morgan virus) is mainly transmitted by an ectoparasitic swallow bug (*Oeciacus vicarious*). The main known avian reservoirs are two passerines, colonially nesting American cliff swallow (*Petrochelidon pyrrhonota*) and house sparrow, that nest in cliff swallow colonies. BCRV can cause encephalitic infections, especially in chicks [230], and selectively infects brain tissue [231]. The prevalence of naturally-acquired BCRV is lower in nestling cliff swallows (~2%) than in nestling house sparrows (20%) [232].

To the best of our knowledge, Venezuelan Equine Encephalitis virus (VEEV) and Western Equine Encephalitis virus (WEEV) have not been described in passerines this century and will be omitted in this review. WEEV is better considered as a submerging virus since human cases have declined dramatically [233], though virulence has not. 

*Alphavirus* taxa, with the likely exception of BCRV, have high zoonotic potential, are known to cause mortality and morbidity in humans and other animals, and generally have a wide host range. EEEV and MAYV are described as emerging viruses. It is feared that climate change may affect vector distribution patterns, potentially causing EEEV emergence in new geographical areas [234]. Other *Alphavirus* taxa discussed above are not currently considered emerging viruses. However, they are poorly described, share many characteristics with EEEV, and have the potential to become emerging viruses.

#### 2.5.3. Family *Coronaviridae*

Members of the *Coronaviridae* are enveloped ssRNA spherical viruses, ranging in size from 120 to 160 nm in diameter, with approximately 30 kb genomes. They are found worldwide and infect all classes of vertebrates [235,236] and are notorious for the human respiratory pathogen SARS-CoV-2. Currently, the family is divided into four genera: *Alphacoronavirus* and *Betacoronavirus* which infect mammals; *Gammacoronavirus* and *Deltacoronavirus*; which infect primarily birds and mammals [236]. *Gammacoronavirus* includes Infectious Bronchitis Virus (IBV), an economically important virus that can cause morbidity and mortality in poultry [237,238], though IBV has not been detected in passerines. Coronaviruses have been detected in wild birds from at least ten orders, including Passeriformes like Ploceidae and Sturnidae [239]. All ICTV-accepted passerine coronaviruses are *Deltacoronavirus*, which includes six taxa that infect birds, and one that infects pigs (Coronavirus HKU15). Coronavirus was first isolated from passerines in 2009 in Hong Kong in a study of 1541 deceased passerine birds; 21 individuals were positive including gray-backed thrush *(Turdus hortulorum*), common blackbird, white-rumped munia (*Lonchura striata*), red-whiskered bulbul (*Pycnonotus jocosus*), and Chinese bulbul (*Pycnonotus sinensis*), with an overall prevalence of 1.3% [240]. Coronaviruses have also been recovered from Japanese white-eye (*Zosterops japonicus*) and Eurasian tree sparrow (*Passer montanus*) [241]. Coronavirus prevalence in passerines seems relatively low. Additional studies have found a prevalence of 0 and 0.9% in passerines in Poland and Kazakhstan, respectively, across 8 and 11 families [242,243].

While the prevalence of coronaviruses in passerines seems relatively low, there is some suggestion that transmission of coronaviruses can occur between poultry and wild birds, and among wild bird populations. It is likely that migrating birds introduce new IBV variants to different regions, though the number of new strain introductions is limited, suggesting that wild birds are probably poor vectors [244]. Nonetheless, coronavirus transmission between poultry and wild birds risks the generation of new recombinant viruses, considering the ability of coronaviruses to infect multiple hosts [245]. 

Coronaviruses circulate in wild passerines, though it is unknown if they cause disease, and if they deserve to be considered “emerging”. Evidence of emergence in passerines is limited, though the prospect of novel coronavirus diseases and epidemics in animals and humans is highly concerning. Continued surveillance is crucial to control and prevent future epidemics. This should focus on coronavirus transmission dynamics, particularly in wild bird populations and their potential zoonotic risk. By gaining a better understanding of these mechanisms, proactive measures can be implemented to mitigate the risks associated with emergence.

### 2.6. Non-Enveloped Positive-Sense ssRNA Viruses

#### 2.6.1. Family *Picornaviridae*


*Picornaviridae* is a diverse family of ssRNA small viruses, with a genome of about 6.7–9.8 kb in size enclosed in icosahedral capsids of about 22–30 nm in diameter. Picornaviruses are known for their remarkable stability, which allows them to survive in extreme environmental conditions, a basis for their easy transmission among animals. This facilitates their transmission via diverse mechanisms, including the fecal-oral route, direct contact with infected birds, and possibly through invertebrate vectors. Field strains of picornaviruses are mainly found in the digestive tract, and mostly cause avian encephalomyelitis in young chickens, pheasants, quail, and turkeys [246]. 

The *Picornaviridae* family currently consists of 158 species grouped into 68 genera, at least 15 of them affecting different avian species. Recent studies using sequence and viral metagenomics analysis have identified and grouped several picornaviruses in the genera *Oscivirus, Passerivirus and Poecivirus* which affect passerine species. 

The *Passerivirus* A1 genome was obtained from pale thrush (*Turdus palidus*), and partial sequences were recovered also from one common blackbird and five gray-backed thrush [247]. The same study also yielded sequences for *Oscivirus* A1 and *Oscivirus* A2 from Muscicapidae and Turdidae, respectively. This study used the taxonomy *Turdivirus- 1, 2* and *3* (TV1, and TV2), though this has now been reclassified (see Table 1). The *Passerivirus* B genome, a potential finch picornavirus, was recently recovered from a diseased violet-eared waxbill (*Granatina granatina*) [248]. This taxon was recovered from a gastroenteric outbreak that occurred 7 to 10 days after violet-eared waxbill and purple grenadier (*G. ianthinogaster*) were introduced into a small breeding aviary in Hungary in 2014, probably because of a neglected quarantine procedure. The aviary population comprised a total of 54 birds from five different Estrildidae genera. Of note, this outbreak had a high mortality rate (65.4%). High viral loads were found in feces, spinal cord, and lungs. The necropsy of two primary affected finches showed clear signs of hepatosplenomegaly and enteritis, suggesting a generalized infection, for which the passerivirus might be a causative agent. 

Avian keratin disorder (AKD) is a disease that causes debilitating beak overgrowth, often with crossing and marked curvature, in many avian species worldwide. It was first documented in black-capped chickadee (*Poecile atricapillus*) in Alaska during the late 1990s. Its average prevalence was 6.5% in adults, and affected individuals showed a significant decline in fitness and survival likely resulting from beak deformities [249,250]. A novel picornavirus, genus *Poecivirus*, was identified using metagenomic sequencing of beak tissue visibly affected by AKD from black-capped chickadee (BCCH) [251]. Screening using PCR and Sanger sequencing of 28 BCCH revealed that 100% of individuals with clinical signs of AKD tested positive for poecivirus, whereas only 9.4% of asymptomatic individuals did. Moreover, this virus was found actively replicating in the beak tissue of AKD-affected BCCH, suggesting a fecal-oral route for its transmission, including poultry. Poecivirus was also present in six additional avian species, which also showed AKD-consistent deformities that were found more broadly through North America and not geographically restricted to Alaska [252], and possibly other continents [249]. 

Recently, a complete genome of a novel French Guiana Picornavirus (FGPV) was detected for the first time in the cloacal sample of a rufous-throated antbird (*Gymnopithys rufigula*) belonging to the large passerine bird family of *Thamnophilidae* [13]. Phylogenetic analysis of the complete genome placed FGPV in a divergent branch within *Picornaviridae* family, sister to the clade grouping the genera of viruses only detected in avian species as *Avihepatovirus*, *Avisivirus, Aalivirus* and *Orivirus*. The bird which tested positive for FGPV, was coinfected with an astrovirus. However, due to the typically asymptomatic nature of picornavirus infections, it is challenging to assess the specific impact of FGPV.

#### 2.6.2. Family *Astroviridae*

Astroviruses are small icosahedral viruses (around 35 nm in diameter) with ssRNA genome, 6.4–7.7 kb in size. They have been classified into two genera, namely *Mamastroviru*s (MAstVs), and *Avastrovirus* (AAstVs) known to infect mammalian and avian species, respectively.

Astroviruses, are typically found in the gastrointestinal tract and are primarily transmitted fecal-orally, as they may persist in the environment, particularly in fecal matter, causing enteric infection in many animal species including humans [253]. Astrovirus infections usually occur asymptomatically in birds, only causing mild disease. Moreover, AAstVs are often detected in birds alongside other viral pathogens and these co-infections can potentially influence the clinical outcomes and severity of disease [254,255]. Environmental factors, such as temperature and humidity, can influence viral survival and transmission dynamics [256]. 

Astroviruses exhibit considerable genetic diversity, with multiple distinct genotypes identified in AAstVs that can be further classified into three principal monophyletic clades related to the three recognized species *Avastrovirus 1*, *2* and *3* [257]. Although astrovirus diversity remains largely unexplored in passerines, large-scale molecular epidemiological studies, coupled with viral sequencing and phylogenetic analyses, improve the understanding of astrovirus diversity and evolutionary dynamics in passerines. Initial studies have detected astroviruses in diverse passerine families, including Fringillidae, Parulidae, and Passeridae [258,259]. The first detection of an astrovirus in passerines was from a black-naped monarch (*Hypothymis azurea*) in Cambodia [260]. Phylogenetic reconstruction and pairwise distance analysis indicated that this AAstV was a novel lineage, potentially AAstV-4, and is well differentiated from previously described AAstVs. Four complete astrovirus genomes (PasAstV1-4) were recovered from cloacal swabs from 50 Neotropical birds collected from the French Guiana rain forest using viral metagenomics studies [258]. A comparison of both the ORF1b and ORF2 amino-acid sequences and genetic distance analysis concluded that the viral sequences identified in these passerines are representative of a putative AAstV group 5, according to the ICTV classification criteria. Of note, the closest relative to all four PasAstV in databases was the only one passerine astrovirus described in a black-naped monarch, with 59–63% amino acid identity in ORF1b. 

The discovery of these astrovirus species raises the possibility that additional astrovirus lineages may exist in the Passeriformes. This also highlights the potential for astroviruses to circulate silently within passerine populations and domestic birds, through a fecal-oral cycle [261,262]. There is ample evidence that cross-species transmission can occur and that individual species may carry divergent astrovirus strains, indicating their sensitivity to infection [263,264,265,266]. 

#### 2.6.3. Family *Caliciviridae*

Caliciviruses are viruses with a linear RNA genome of 6.4–8.5 kb. The family comprises 11 genera, two of which are exclusively avian viruses, *Bavovirus* and *Nacovirus*, identified recently in chickens and turkeys in Northern Europe, America and Eurasia [267,268,269,270]. Recent metagenomic studies have recovered novel calicivirus genomes in diverse avian hosts, in some cases revealing high calicivirus diversity and suggesting the existence of new genera [10,15,271,272,273]. Avian caliciviruses identified in diverse avian species show either no sign of infection or present with gastroenteritis, poor feather condition and/or infectious runting-stunting syndrome [274]. 

Calicivirus has been identified from several passerines. Metagenomic and similar analyses have shown that caliciviruses may also infect invertebrates, and on several occasions high identity has been shown between caliciviruses from passerines and from arthropods, suggesting a potential cross-group cycle [275]. A calicivirus isolated from tomtit (*Petroica macrocephala*) shared >99% of amino acid identity with a calicivirus library isolated in woodlice (*Isopoda* sp.), highlighting a potential food web interaction [275]. In a similar metatranscriptomic study [7] calicivirus-like viruses, were obtained from three blackbirds and one dunnock (*Prunella modularis*). Two of the three blackbird viruses were closely related to bee-infecting viruses. Metatranscriptomic analysis of the virome of the invasive Indian myna (*Acridotheres tristis*) in Australia [276] recovered two calicivirus fragments in samples from the brain, liver, and large intestine. These novel caliciviruses shared less than 80% identity with their closest relatives from other avian species. Finally, a recent study from China recovered two novel *Calicivirus* genera, tentatively designated *Zhenovirus* and *Birmovirus*, from four passerine species (out of 49): *Phoenicurus auroreus*, *Coccothraustes coccothraustes*, *Emberiza chrysophrys*, and *Prunella montanella* [15] with no clinical signs. This suggests that the viruses may not cause noticeable clinical signs in their avian hosts.

Caliciviruses have only been known from birds for about 10 years, and there is very little evidence to demonstrate whether they are emerging viruses, or whether they cause disease in passerines or other wild bird groups, though the presence of relatively minor disease correlates with calicivirus-positive domestic birds. However, as novel and diverse caliciviruses are increasingly being recovered from wild passerines, the taxonomy and distribution of this group require further investigation.

#### 2.6.4. Family *Hepeviridae*

*Hepeviridae* capsids are small (around 32–34 nm in diameter) and icosahedral, and sometimes enveloped (as when isolated from blood or in tissue culture), but naked when in feces or in bile. The genome is around 7.2 kb. Hepeviruses are widely represented in vertebrates and invertebrates. Well-known taxa include both human and avian hepatitis E. The latter can cause hepatitis-splenomegaly syndrome in chickens. Additionally, diverse hepe-like viruses have been described from metagenomic analyses of vertebrates, including birds in China [15], French Guiana [13] and New Zealand [7]. The hepe-like viruses identified in several passerines, including silvereye (*Zosterops lateralis*), song thrush [7] and rufous-throated antbird [13] could be passerine hepeviruses. However, further investigation is required to ascertain that these are avian viruses, and not invertebrate hepe-like viruses, recovered from insectivorous birds [275]. Surprisingly, a 2–10% seroprevalence against Hepatitis E virus (HEV) has been described from three pet bird species from the families Cardinalidae, Alaudidae and Fringillidae, in northeast China [277].

### 2.7. Double-Stranded RNA (dsRNA) Viruses

#### Family *Reoviridae*


Genus *Rotavirus* (RV) from the subfamily *Sedoreovirinae* are non-enveloped double-stranded RNA viruses, with a genome composed of 11 segments, adding up to 18.5 kbp, that encode six structural and five non-structural proteins [278]. The genus is classified into nine species A–J [279]. RVA is the most commonly detected in humans and animals, while RVD, RVF, and RVG have been isolated exclusively in poultry [280]. RV is a leading cause of gastroenteritis, and well-described clinical signs include diarrhea, dehydration, anorexia, and weight loss in mammals, and high mortality in birds [281]. Avian rotaviruses (AvRV) have been detected in a wide variety of birds including passerines [282].

RVA has been detected in the widespread reed bunting (*Emberiza schoeniclus*) in Hungary [283]. Phylogenetic analysis based on partial VP6 sequence suggests a close genetic relationship with turkey rotaviruses. RVA was also detected in Brazil in two species of *Turdus*, domestic canary (*Serinus canaria* var *domestica*), and chestnut-bellied seed-finch (*Sporophila angolensis*) [284]. Interestingly, the results of sequence analysis revealed the genotypes G3/P (Turdus sp.), G3 (*Serinus canary*), G1/P (*Turdus rufiventris*), and G6 (*Sporophila angolensis*). Additionally, the nucleotide sequences in this study showed more similarity to human strains than to the wild bird strains [285].

As well as posing a threat to poultry and a wide range of wild birds, it has been suggested that rotaviruses may have zoonotic potential. RVA genotype G3, which is considered the third most prevalent genotype has a very broad host range, comprising artiodactyls, carnivores, chiropterans, leporids, perissodactyls, rodents, simians (including humans), and passerines [285]. Similarly, RVA genotypes G1 and G6 have been documented in a wide host range [284]. Some authors have suggested that interaction between humans and wild passerines, potentially due to environmental disturbances [286], or through illegal commercialization, can lead to spillover of rotaviruses to new hosts [287]. 

Due to their broad host range, and their negative consequences on poultry AvRVs are considered emerging pathogens [286] and strikingly, there is some suggestion that they may be zoonotic. RVA incidence in birds is generally low [288], and there have been few studies on RVA in wild passerines to support these suggestions.

### 2.8. RNA Reverse Transcribing Virus

#### Family *Retroviridae*

Retroviruses are enveloped viruses (about 100 nm in diameter) whose genome consists of two identical molecules of ssRNA (7–10 Kb in length), which is reverse transcribed into dsDNA and integrated into the genome to form the provirus. Avian leukosis virus (ALV) is placed in the genus *Alpharetrovirus* in the family *Retroviridae*. ALV is mostly found in Galliformes and, recently, in Anseriformes. The few attempts to detect ALV in Passeriformes mostly failed or did not present convincing evidence of ALV infection. The most worrying subgroup since its emergence in the 80s, ALV-J, has rapidly extended its host range, affecting other avian orders as well as Galliformes. Cross-species infection has been shown when Galliformes are housed together with species from other orders [289], and novel ALV-J strains have been detected in Anseriformes [290]. 

Lymphoid leukosis was recorded from captive passerines (*Tangara schrankii* and Pterorhinus *albogularis*) based on gross post-mortem and histopathology findings [286], but as the aetiological agent was not identified it cannot be confirmed that it was ALV. Antibodies to the closely related Rous Sarcoma virus (RSV) were detected from a pool of three house sparrows in the Czech Republic using neutralization tests [291], though the authors could not conclude that sparrows participate in ASLV transmission. ALV-J sequences were recovered from 2/129 (1.6%) passerines (*Emberiza elegans* and *Tarsiger cyanurus*) in China [292]. A second study from China detected ALV-J in 4/315 (1.3%) passerine samples (from *Poecile palustris*, *Phylloscopus inornatus*, and again from *T. cyanurus* and *E. elegans)* [293]. Sequences showed relatively low identity (<75%) with other known ALV-J strains but the 3′UTR region was 98% identical. Surprisingly, deletions found in the 3’UTR region in wild-bird isolates were similar to those in Chinese layer chicken isolates, indicating that ALV-J found in passerine birds and poultry may have a common origin [293]. None of these six passerine individuals showed signs of pathology during necropsy. 

Although there are signs of ALV infection in passerines, which might be attributable to ALV-J plasticity increasing its host range, further research is needed to determine if these are rare cases or if this is generally happening around the world undetected. Evidence so far suggests that ALV infection in passerines might be asymptomatic.

## 3. General Recommendations to Reduce the Risk of Transmission of Viruses Associated with Passerines

Although most virus-infected passerines probably do not pose a serious risk to healthy people, it is necessary for everyone interacting with these birds to observe appropriate biosecurity measures to decrease the spread of viruses to other birds, animals, or the environment and reduce the risk of transmission of potentially zoonotic pathogens. General recommendations for handling passerines are similar to those for other birds (pets, poultry or wild birds): (a) avoid direct contact with or touching sick or dead birds found in the field, including bird feces and feathers; (b) wear gloves if it is necessary to pick up birds, for example, to take it to a veterinary center; (c) wash hands thoroughly after handling the bird and any surfaces in contact with it. 

Professionals (such as veterinarians, biologists, technicians, and ornithologists) must use appropriate personal protective equipment (PPE: surgical mask, safety glasses, protective gloves, laboratory coat) when collecting samples from passerines for laboratory analysis or in case of post-mortem examination even if there are no obvious signs of disease. If necessary, a level 2 or 3 biosafety cabinet should be used.

It is very important that farms, backyard flocks and passerine breeding facilities implement measures to discourage access by wild passerines (such as sparrows, or starlings) and reduce the spread of pathogens (since transmission may be a two-way route between indoor and outdoor birds). These measures include (a) strict biosecurity measures (restrict access only to authorized personnel and vehicles, use of exclusive clothing and footwear for handling birds, etc.); (b) adequate cleaning and disinfection protocols (for equipment, clothing, vehicles, facilities, etc.); (c) keep food and water equipment indoors or in closed areas, cover ponds in netting, close windows and doors completely, cover holes in ceilings and walls, etc.

Finally, official health agencies must be informed when it is suspected that a passerine may be affected by a notifiable disease such as avian influenza or West Nile fever.

## 4. Conclusions

Several significant viruses have emerged in recent years associated with passerine birds. Some have caused significant mortality in wild and domestic birds, including economically important poultry. Others have spilled over into mammals, and some, especially the flaviviruses, have caused significant human morbidity and mortality. In some cases, passerines are critical to these viral transmission cycles. In others, passerines may be a bridge between natural virus reservoirs and poultry and/or human populations, though much more investigation is needed to fully understand this possibility. Therefore, there is a growing interest in documenting the viruses that circulate in passerine birds.

Much of the data about new and emerging viruses in birds comes from metagenomic analyses. However, the investigation of the global virome is still in its infancy, and investigation of the Passeriformes virome is not a priority. Passeriformes are the most diverse avian order, and many passerine species are extremely abundant, giving them critical mass to amplify virus transmission cycles. Others undergo annual migrations and can potentially introduce new agents to different continents. Many passerines live in anthropomorphic settings, such as cities and farms, and potentially contribute to spillover. The attentive reader of this review will have noticed that the same passerine families and species frequently recur (American crow, common blackbird, great tit, house sparrow, zebra finch, etc.). This highlights that our understanding of the Passeriformes virome is weighted towards abundant, easy-to-sample species, plausibly including those most susceptible to illness and death. But even the best-investigated virus taxa have only been detected in a small fraction of Passeriformes species. West Nile virus, for example, has been recorded in about 156 passerine species, which is just 2.5% of Passeriformes. It is likely that no virus has been recovered from the majority of passerine species, and that most passerine species have never been sampled. There seem to be clear biogeographical patterns in some virus families, notably for *Flaviviridae* and *Togaviridae*. However, it is often difficult to discern if there is a pattern as the sampling effort is very patchy and dominated by rich countries. The virome is best described from Europe, North America, Australia/New Zealand and increasingly from Asia. However, global trade, current and historic, appears to have introduced viral taxa to new continents: the location of virus detection and the area where the virus evolved may not be identical.

A diverse and growing range of viruses have been detected in passerine birds (Table 1), and it would appear that several are “emerging viruses” if we consider an increase in their incidence, their expansion to new geographic areas, new hosts, new viral lineages, and increased virulence (Figure 4). 

Some of the increase in “virus emergence” may be explained by increased awareness of new viruses as novel tools (i.e., metagenomics) have also emerged that facilitate the discovery of new viruses. Caution is required in discussing viruses discovered in the absence of clinical signs of infection, or clarity about the identity of host species. Increasing discovery of novel viruses is not a synonym for viral emergence. However, it is also difficult to identify virus emergence because the baseline is so poorly defined. 

There is ample evidence that viruses of animals can cause huge problems to human health and to the economy but can also have an impact on biodiversity and ecosystems balance. It is extremely important that we evaluate the risk by investigating the viral diversity in wild animal hosts. And much more research is required.

**Table 1 microorganisms-11-02355-t001:** Virus names, abbreviations and genus/species designations are listed according to the taxonomy and nomenclature approved by the International Committee on Taxonomy of Viruses (ICTV) [294]. All ICTV accepted viral taxa are in italics.

Viral Family	Genus	Viral Taxa	Abbreviations	Nomenclature	Bird Family	Clinical Signs	Mortality Rate	References
**dsDNA VIRUSES**								
** *Herpesviridae* **	*Iltovirus*	*Gallid herpesvirus 1*	GaHV-1	*Iltovirus gallidalpha1*	Turdidae, Estrildidae, Sturnidae	Subclinical infection/ Respiratory signs	High	[7,26,34,35,37]
		*Psittacid herpesvirus 1*	PsHV-1	*Iltovirus psittacidalpha1*	Sturnidae			
		passerid herpesvirus 1		not recognized	Fringillidae			
	*Mardivirus*	*Columbid herpesvirus 1*	CoHV-1	*Mardivirus columbidalpha1*	Corvidae, Turdidae	Subclinical infection	Probably not	[40]
** *Poxviridae* **	*Avipoxvirus*	*Fowlpox virus* (Lineage A1)	FWPV	same	Fringillidae, Passeridae	Cutaneous lesions/Upper respiratory and digestive tract lesions	High	[42,43,44,45,46,47,48,49,50,51,52,53,54]
		*Pigeonpox virus* (Lineage A2)	PGPV	same	Fringillidae		High/Medium	
		Accipitriformespox virus (Lineage A7)		same	Lanidae		High/Medium	
		*Canarypox virus* (Lineage B1)	CNPV	same	Fringillidae, Paridae, Corvidae (and 11 other families)		High/Medium	
		*Starlingpox virus (Lineage B2)*		same	Sturnidae, Passeridae		High/Medium	
		Lineage B3 poxvirus		not recognized	Corvidae (and eight other mostly Passeriformes families)		High	
** *Adenoviridae* **	*Atadenovirus*	European robin adenovirus		not recognized	Meliphagidae, Fringillidae, Ploeceidae, Estrildidae, Turdidae, Ptilonorhynchidae, Sturnidae, Zosteropidae, Maluridae, Petroicidae	Subclinical infection/ Not known	Probably not	[57,58]
		European greenfinch adenovirus		not recognized				
		Eurasian siskin adenovirus		not recognized				
		Eurasian bullfinch adenovirus		not recognized				
		common chaffinch adenovirus		not recognized				
		passerine adenovirus 1		not recognized				
		white plumed honeyeater adenovirus 1, -2		not recognized				
		vitelline masked weaver adenovirus strains 37869 and 38132		not recognized				
	*Aviadenovirus*	great tit adenovirus strain 47292		not recognized	Fringillidae, Estrildidae, Paridae, Ploceidae, Sylvidae, Falcunculidae, Petroicidae	Subclinical infection/ Not known	Probably not	
		european greenfinch adenovirus		not recognized				
		goldfinch adenovirus		not recognized				
		vitelline masked weaver adenovirus strain 39658		not recognized				
	*Siadenovirus*	double-barred finch adenovirus		not recognized	Estrildidae, Paridae, Sylvidae, Thraupidae, Meliphagidae, Pardalotidae, Rhipiduridae, Sturnidae	Subclinical infection/ Probable renal lesions	Probably not	[56,60]
		eurasian blackcap adenovirus		not recognized				
		gouldian finch adenovirus 1	GFAdV-1	not recognized				
		great tit adenovirus strain 47292	GTAdV-1	*Great tit siadenovirus A*				
		zebra finch adenovirus strain 47535		not recognized				
** *Papillomaviridae* **	*Etapapillomavirus*	*Fringilla coelebs Papillomavirus 1*	FcPV1	*Etapapillomavirus 1*	Fringillidae	Cutaneous lesions	Low	[66,67,69,70,71,72]
		Serinus canaria papillomavirus 1	ScPV1	not recognized	Fringillidae	Not known	Not known	[68]
**ssDNA VIRUSES**								
** *Circoviridae* **	*Circovirus*	*Beak and feather disease virus*	BFDV	*Circovirus parrot*	Artamidae, Corvidae, Estrildidae, Fringillidae, Petoicidae, Sturnidae, Thamnophilidae	Feather lesions	Probably not	[76,77,78]
		*Circovirus canary*	CaCV	same		Feather diseases and immunosupression/ Subclinical infection	Yes/Not known	[25,79,80,81,82]
		*Circovirus finch*	FiCV	same				
		*Circovirus raven*	RaCV	same				
		*Circovirus starling*	StCV	same				
		*Circovirus zebrafinch*	ZfiCV	same				
	*Cyclovirus*	*Robinz virus RP_736*	RobinzV736	*Cyclovirus cervienka*	Petroicidae	Not known	Not known	[17]
		*Robinz virus RP_1170*	RobinzV1170	*Cyclovirus liepsnele*				
		*Robinz virus RP_493*	RobinzV493	*Cyclovirus pettirosso*				
		*Robinz virus RP_620*	RobinzV620	*Cyclovirus prihor*				
		*Robinz virus RP_526*	RobinzV526	*Cyclovirus punarinta*				
		*Robinz virus RP_584*	RobinzV584	*Cyclovirus rudzik*				
		*Robinz virus RP_259*	RobinzV259	*Cyclovirus totoi*				
	Probably new genus	Gyrovirus 11		not recognized	Thamnophilidae	Not known	Not known	[89]
** *Genomoviridae* **	*Gemycirculavirus*	new species		not recognized	Petroicidae, Aegithalidae, Corvidae, Emberizidae, Fringillidae, Muscicapidae, Paridae, Sylviidae, Turdidae, Zosteropidae	Not known	Not known	[88]
		*Chickadee associated gemycircularvirus 1*		*Gemycircularvirus chickad1*	Paridae			[85]
		Finch associated genomovirus 5		*Gemycircularvirus haeme1*	Fringillidae			[86]
		Finch associated genomovirus 6		*Gemycircularvirus haeme2*				[86]
	*Gemykibivirus*	*Blackbird associated gemykibivirus 1*		*Gemykibivirus blabi1*	Turdidae			[87]
		*Black robin associated gemykibivirus 1*		*Gemykibivirus blaro1*	Petroicidae			
		Finch associated genomovirus 3		*Gemykibivirus haeme1*	Fringillidae			[86]
		Finch associated genomovirus 2		*Gemykibivirus haeme3*				
		Finch associated genomovirus 4		*Gemykibivirus haeme4*				
		Finch associated genomovirus 1		*Gemykibivirus haeme5*				
	*Gemykroznavirus*	Finch associated genomovirus 8		*Gemykronzavirus haeme1*	Fringillidae			
	*Gemygorvirus*	*Starling associated gemygorvirus 1*		*Gemygorvirus stara1*	Sturnidae			[87]
** *Parvoviridae* **	*Aveparvovirus*	*Aveparvovirus passeriform1*	PfPV	same	Thraupidae	Not known	Not known	[95]
**ssRNA VIRUSES**								
** *Orthomyxoviridae* **	*Alphainfluenzavirus* ^1^	Highly Pathogenic Avian Influenza H5N1	FLUAV/(H5N1)	*Alphainfluenzavirus influenzae*	Passeridae, Turdidae, Ploceidae, Corvidae, Hirundidae, Icteridae (and many other families)	Mild/Severe disease	High	[101,102,103,104,105,106,107,108,110,116]
** *Paramyxoviridae* **	*Orthoavulavirus*	*Avian paramyxovirus 1* (NDV, *Newcastle Disease virus*)	APMV-1	*Orthoavulavirus javaense*	Passeridae, Motacillidae, Passarellidae	Not known/Subclinical disease	High/Not known	[113,123,124,126,127]
	*Metaavulavirus*	Avian paramyxovirus 2	APMV-2	*Metaavulavirus yucaipaense*	Corvidae, Motacillidae, Troglodytidae, Muscicapidae, Emberizidae, Estrildidae	Mild/Severe disease		[128,129,130,131,132,133,134]
	*Paraavulavirus*	Avian paramyxovirus 3	APMV-3	*Paraavulavirus wisconsinense*				[135,136]
	*Paraavulavirus*	Avian paramyxovirus 4	APMV-4	*Paraavulavirus hongkongense*				[138]
** *Pneumoviridae* **	*Metapneumovirus*	*Avian metapneumovirus*	AMPV	*Metapneumovirus avis*	Passeridae, Sturnidae	Not known	Not known	[141,142]
** *Bornaviridae* **	*Orthobornavirus*	*Estrildid finch bornavirus 1*	EsBV-1	*Orthobornavirus estrildidae*	Estrildidae, Corvidae, Fringillidae	Not known	Not known	[146,147,148,149]
		*Canary bornavirus 1*	CnBV-1	*Orthobornavirus serini*				
		*Canary bornavirus 2*	CnBV-2					
		*Canary bornavirus 3*	CnBV-3					
** *Flaviviridae* **	*Orthoflavivirus* ^1^	*Japanese encephalitis virus*	JEV	*Orthoflavivirus japonicum*	Passeridae, Corvidae, Fringillidae, Turdidae (and >23 more families)	Not known	High	[183]
		*Saint Louis encephalitis virus*	SLEV	*Orthoflavivirus louisense*				[191,192,193,194]
		*Murray Valley Encephalitis virus*	MVEV	*Orthoflavivirus murrayense*				[186]
		*Usutu virus*	USUV	*Orthoflavivirus usutuense*		Severe disease		[174,175,176]
		*West Nile virus*	WNV	*Orthoflavivirus nilense*		Mild/Severe disease		[158,159,160,161,162,169]
** *Togaviridae* **	*Alphavirus* ^1^	*Eastern equine encephalitis virus*	EEEV	same	Turdidae, Hirundidae, Icteridae, Petroicidae, Estrildidae, Monarchidae	Neurological disease	Yes	[206,207,208]
		*Highlands J virus*	HJV	same		Mild disease		[210]
		*Sindbis virus*	SINV	same		Severe disease		[215,216]
		*Ross River virus*	RRV	same		Not known		[219,220]
		*Mayaro virus*	MAYV	same				[225,226,227,228]
		*Buggy Creek virus*	BUCGV	*Fort Morgan virus*		Neurological disease		[230,231,232]
	*Deltacoronavirus* ^2^	*Bulbul coronavirus HKU11*	BulCV_HKU11	same	Turdidae, Estrildidae, Pycnonotidae, Passeridae, Zosteropidae, Fringillidae, Emeberizidae	Severe disease	Probably high	[239,240,241]
		*Munia coronavirus HKU13*	MunCV_HKU13	same				
		*Thrush coronavirus*		not recognized				
		*White-eye coronavirus HKU16*	WECoV_HKU16	same				
		*Coronavirus HKU15*	PoCoV_HKU15	same				
** *Picornaviridae* **	*Oscivirus*	Oscivirus A1 and A2 (turdivirus 2 and 3)	OsV-A1/OsV-A2	*Oscivirus A*	Turdidae, Muscicapidae	Severe disease	Not known	[247,248]
	*Passerivirus*	*Passerivirus A and B*	PasV-A/PasV-B	same	Turdidae, Estrildidae		High	
	*Poecivirus*	*Poecivirus A1*	PoeV-A	*Poecivirus A*	Paridae		Yes	[251,252]
	Probably new genus	French Guiana Picornavirus	FGPV	not recognized	Thamnophilidae	Not known	Not known	[13]
** *Astroviridae* **	*Avastovirus*	novel avastrovirus clades 4 and 5		not recognized	Fringillidae, Monarchidae Parulidae, Passeridae	Subclinical infectionMild disease	Not known	[258,259,260]
** *Caliciviridae* **	Probably new genera	new calicivirus		not recognized	Prunellidae, Turdidae, Emberizidae, Fringillidae, Muscicapidae, Petroicidae	Subclinical infection	Not known	[7,15,275,276]
** *Hepeviridae* **	Probably new genera	new hepe-like viruses		not recognized	Thamnophilidae, Turdidae, Zosteropidae	Not known	Not known	[7,13,15]
**dsRNA VIRUSES**								
** *Reoviridae* **	*Orthoreovirus* ^2^	new orthoreovirus		not recognized	Emberizidae, Turdidae, Fringillidae	Not known	High mortality	[283,284]
**RNA REVERSE TRANSCRIBING VIRUS**								
** *Retroviridae* **	*Alpharetrovirus*	*Avian leukosis virus*	ALV	same	Emberizidae, Paridae, Corvidae, Muscicapidae, Thraupidae, Phylloscopidae	Subclinical infection	Possibly	[291,292,293]

^1^: Zoonotic risk; ^2^: Doubtful zoonotic risk.

## Figures and Tables

**Figure 1 microorganisms-11-02355-f001:**
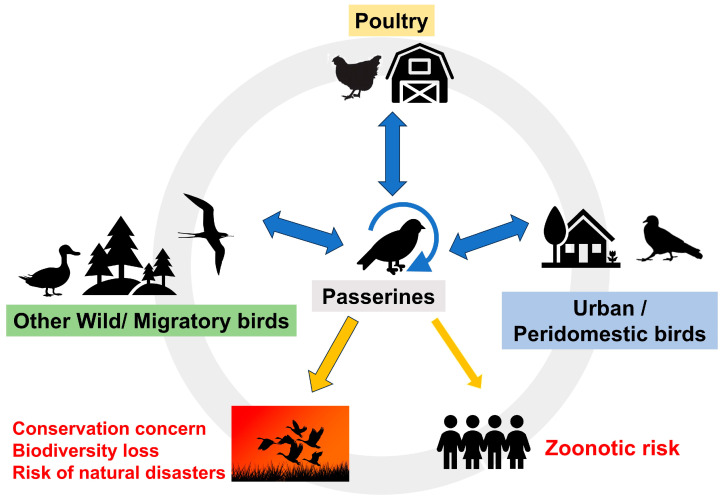
The theoretical role of passerine birds in the circulation and potential spillover of viruses and other infectious agents to other passerines (blue semi-circular arrow around the bird in the center), domestic, peridomestic and other wild birds (blue bidirectional arrows) and their relation to potential zoonotic risk or threats to biodiversity (orange unidirectional arrows). The relation of passerines to theoretical consequences, such as zoonotic risk or threats to biodiversity, are depicted using orange unidirectional arrows. The One Health concept is represented by the light grey ring.

**Figure 2 microorganisms-11-02355-f002:**
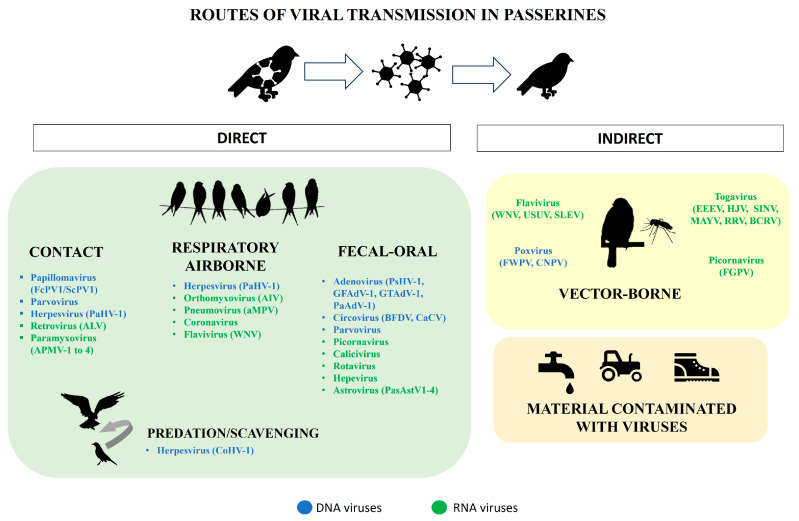
Viral transmission routes in passeriform birds. DNA viruses are shown in blue and RNA viruses in green. Avian Influenza virus (AIV), Avian leukosis virus (ALV), APMV (Avian paramyxovirus), APV (Avian papillomavirus), BCRV (Buggy Creek virus), BFDV (Circovirus), CaCV (Circovirus), CNPV (Canary poxvirus), CoHV-1 (Columbid herpesvirus-1), CoV (coronavirus), HEV (hepevirus), CRESS-DNA viruses (circular-rep encoded single stranded DNA), CV (calicivirus), EEEV (Equine eastern encephalitis virus), Estrildidae adenovirus (EsAdV), FcPV (Fringilla coelebs papillomavirus), FGPV (Picornavirus), FWPV (Fowlpox virus), GFAdV-1 (Gouldian finch adenovirus-1), GTAdV-1 (great tit adenovirus-1), HPAI (Highly pathogenic Avian Influenza), HJV (Highland J virus), MAYV (Mayaro virus), PaHV-1 (Passerid herpesvirus-1), PaPV (Passerid parvovirus), PasAstV (Passerid astrovirus), Passerid adenovirus-1 (PaAdV-1), PsHV-1 (Psittacid herpesvirus-1), RRV (Ross River virus), RVA (rotavirus), ScPV1 (Serinus canaria papillomavirus), SINV (Sindbis virus), SLEV (Saint Louis Encephalitis virus), USUV (Usutu virus), and WNV (West Nile virus).

**Figure 3 microorganisms-11-02355-f003:**
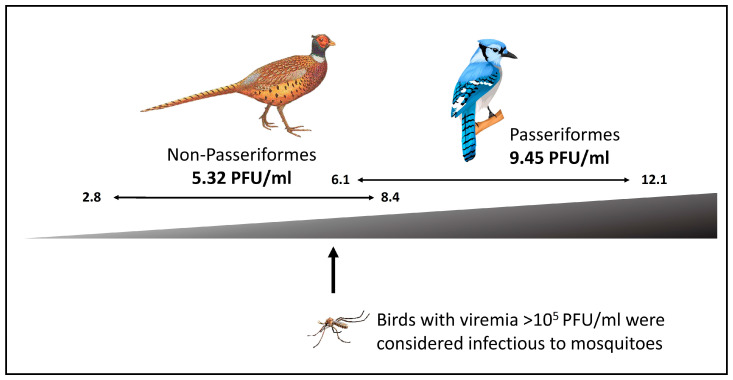
Average peak viremia induced in 15 non-Passeriformes (*N* = 38) and 10 Passeriformes (*N* = 49) species experimentally infected with the NY-99 strain of West Nile virus (WNV). Viremias are shown as log_10_ [147].

**Figure 4 microorganisms-11-02355-f004:**
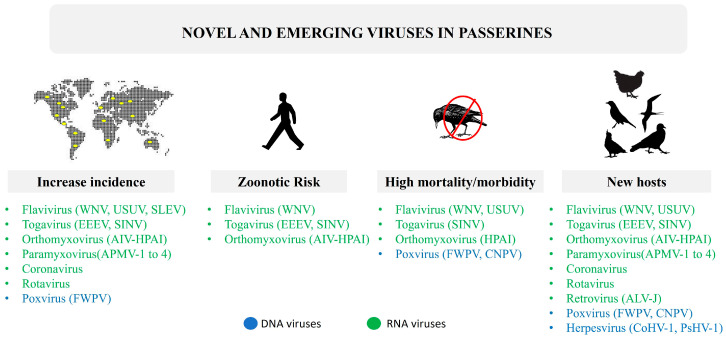
Emerging features of viruses in passeriform birds. Some examples of viruses that affect passerines are shown in parentheses. AIV (Avian influenza virus), APMV (Avian paramyxovirus), ALV-J (Avian Leukosis virus), CNPV (Canary poxvirus), CRESS-DNA viruses (circular-rep encoded single stranded DNA), CoHV-1 (Columbid herpesvirus-1), CoV (coronavirus), EEEV (Equine eastern encephalitis virus), FWPV (Fowlpox virus), HPAI (Highly pathogenic Avian Influenza), PsHV-1 (Psittacid herpesvirus-1), RVA (rotavirus), SINV (Sindbis virus), SLEV (Saint Louis Encephalitis virus), USUV (Usutu virus), WNV (West Nile virus).

## Data Availability

All data used in this review is already publicly available. We provide four Appendix A with summarized information that may be useful for the reader.

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
