# Peer review of "Emerging and Novel Viruses in Passerine Birds"

_microorganisms, 2023, doi:10.3390/microorganisms11092355_

Round 1

Reviewer 1 Report

Dear editor,

Thank you for giving me the chance to review the manuscript  „Emerging and Novel Viruses in Passerine Birds“ by Williams et al as a review article for Microorganisms. The first author Richard Williams and the senior author Laura Benitez both have many years of experience in the field of pathogen discovery and avian disease epidemiology with a track-record of multiple peer-reviewed articles.

The topic of the manuscript is highly interesting to veterinarians dealing with avian patients in their daily work. It is a comprehensive review article that explores the current state of knowledge regarding the virome of passerine birds. The bulk of the article is devoted to a detailed examination of the various viruses that have been identified in passerine birds, including both well-known pathogens such as avian influenza and West Nile virus, as well as newly discovered viruses such as the recently identified Passerivirus. The authors discuss the pathogenic significance of these viruses, including their potential to cause disease in both birds and humans. Personally, I like the vast amount of detailed information in this article, since it is highly valuable for my daily diagnostic work. However, I feel that the authors overemphasize the potential danger of emerging and zoonotic diseases from passerine birds in this manuscript. Current knowledge concerning the microbiome and virome of exotic and avian species is very  low. The first description of a virus in a host is definitively not enough to state an emerging disease. Science should not be a horror story of worst case szenarios, and therefore 99% or more of newly detected viral agents in a passerine birds will be „virome“ and not harm either the main host or any other species. A balanced review should enable veterinarians and ornithologists to focus on the real hazards and select appropriate biosafety measures based on data on epidemiology, pathogenicity and virulence of their actual patient. Drawing horror szenarios in which every bird is the potential host of the next pandemic virus and therefore should be handled under BSL-4 conditions should be avoided.

In summary, this review provides a comprehensive overview of the current state of knowledge regarding the virome of passerine birds, and highlights the importance of continued research in this area for both public health and conservation. I recommend acceptance to the article „Emerging and Novel Viruses in Passerine Birds“ after minor modifications, as follows:

In some instances, the manuscript is unbalanced concerning the threat of zoonotic and emerging diseases coming from passerine species versus the danger of extinction of some passerine species by human actions. I see the attempt of the auhors to comment on overestimating primary litarature in some occations, but still feel that this review overemphasizes the potential danger of emerging and zoonotic diseases from passerine birds. I do not see a major evidence-based viral threat on humanity coming from passeriformes. Even West-Nile-virus, although undoubtly zoonotic seems to be of rather low importance due to the high Ct values in vertebrates. In contrast, there is reasonable concern of extinction of individual passeriform-species. Please check the following article: Manne, L., Brooks, T. & Pimm, S. Relative risk of extinction of passerine birds on continents and islands . Nature 399, 258–261 (1999). https://doi.org/10.1038/20436. I guess it is worth to include it. However, passeriformes and birds as a group are not in danger of extinction: https://www.iucnredlist.org/resources/summary-statistics. I suggest that the authors try to follow the one health concept, in a balanced way, according to the updated tripartite definition: “One Health is an integrated, unifying approach that aims to sustainably balance and optimize the health of people, animals and ecosystems.” (https://www.who.int/news/item/01-12-2021-tripartite-and-unep-support-ohhlep-s-definition-of-one-health). I think, we as  veterinarians shouldn’t see animals only as patients, but also as mostly healthy part of the ecosystem. Maybe, the authors can find some ornithologists, hunters, foresters or other experts in ecology and passerine biology to discuss their ideas.

I have the feeling that the concept of „emerging disease“ is overstretched.  It is possible that this is a consequential error arising from the trend toward publication-promoting overestimation of hazards in the numerous primary studies. I hope that the authors will have the courage to take a meta-position and, where necessary, to present an epidemiological assessment that contradicts individual primary studies. I suggest to include a clearly-stated and literature-backed definition of what is an „emerging disease“ in the introduction. I feel the authors underrate the „disease“, e.g. lesions, suffering, death, … in their current approach. Afterwards, use an appropriate concept of „emerging disease“ throughout the text.

The concept of „emerging viruses“ should also be reconsidered. For example, the sentence in the abstract: “ But adenoviruses, paramyxoviruses, coronaviruses, retroviruses, poxviruses, and herpesviruses could also be considered as emerging viruses.“ This is not appropriate and not backed by any available epidemiological data. I suggest to rewrite all instances where this overstretching occurs in the current manuscript. I acknowledge that the authors mention this problem in one sentence in the conclusions, however, I think they should be more couraged and don’t hide it behind a legion of citations of primary literature claiming their newly reported virus in species xy hast he potential to be the new mass killer, and therefore needs lots of grant money.

The introduction needs to be ammended by an overview on the major families and genra of birds within the order passeriformes. This will markedly help to increase the understandability fort he non ornithologist reader.

At the end of the 1st subsection, The authors state their hypothesis: „It seems evident that the interface between agriculture and wildlife can be a trigger for the spread of new and emergent viruses among different species of birds, increasing the risk of pathogen spillover from domestic chickens, mainly backyard chickens, to wild birds [21] but probably vice versa as well.  Passerines could play a larger role than previously thought in this interaction, a hypothesis that needs to be further studied.“ This sounds good, but a critical evaluation reveals that this is not the real topic of this review, and it can’t be the aim of a review. Which experimental methods and tests would be appropriate to aim at this interesting hypothesis? I guess, you would need paired samplings of poultry an passerine species surrounding the detection of a new and emerging virus – in a set previously known to be negative. I cannot find information like this somewhere in the manuscript.  I suggest to clearly state a hypothesis and aims for this manuscript, which follow the following guidelines: 1.) an aim needs to be specific, 2.) measurable as being reached or not in your final coclusions, 3.) interesting for the reader, 4.) achievable within this manuscript

Figure 1 is not balanced. Passerine species are presented like a parasite or health hazard. In my eyes, this is not appropriate or based on evidence. I suggest that the arrows should be at least double-headed, or even better, take the passerines out of the center: Passerines are not a hazard.

The statement „Moreover, the suspicion that passerines may be infected with a specific virus is sometimes based only in the detection of antibodies during surveys of wild and domestic birds [23]“ should be rephrased, taking the in most situations retrospective view of serology into account.

Table 1 includes highly valuable data. I should be amended by a column concerning the occurrence of disease or subclinical infection, not only death. Furthermore, the table should include the link to the references, especially concerning disease, death and zoonotic risk. I especially wonder if the references concerning zoonotic potential are evidence-based science or just the opinion of someone. Unfortunately the font and colour scheme of the table in the pdf provided for review prevents the deciphering of all words, and I guess there are missing rows. Then content of table 1 needs to be checked again by the editors or reviewers after revision in an acceptable style and front size.

The depth of basic knowledge concerning the viruses in the first sentence of each paragraph differs markedly in the different subsections. In order to be concise, I suggest to add, shape of virion, size of virion, envelope or not, configuration of genome, single or doublestranded, size of genome, RNA or DNA for every family.

Concerning Circoviridae and related viruses, the paragraph should be amended by one or two more sentences introducing what a CRESS is to the non-expert. Furthermore, follow the general guideline of first spelling out and adding an abbreviation afterwards in brackets, and not vice versa: “Although only a small number of viruses have been described in this group of birds so far, the number of new avian gyroviruses, circoviruses, genomoviruses and diverse single-stranded DNA viruses known as CRESS (Circular Rep-encoding single stranded) DNA viruses [84],….”

Concerning avian influenza, the authors state: „Two large surface glycoproteins, haemagglutinin (H), which is responsible for cell-binding and fusion, and neuraminidase (N), which promotes the release of viral progeny, are highly divergent. 16 H types (H1-H16), and nine N types (N1-N9) have been found in birds [95]. These bird strains of IAV are known as Avian Influenza (AI). Due to reassortment of the segmented influenza genome, which can cause antigenic shift, there are 149 possible H-N combinations, at least, in theory, and each combination is considered a different subtype.“ However, 16 * 9 = 144. The authors shoud defend why they use a theoretical number of 149 combinations throughout the manuscript. Furthermore, the references concerning passeriformes as hosts of influenza viruses as well as important flaviviruses are rather old. I suggest to include some references targeting the recent epidemic in Europe: Günter et al. 2023. Emerg Microbes Infect. 12(2):2231561. Continuous surveillance of potentially zoonotic avian pathogens detects contemporaneous occurrence of highly pathogenic avian influenza viruses (HPAIV H5) and flaviviruses (USUV, WNV) in several wild and captive birds. doi: 10.1080/22221751.2023.2231561. This article highlights, that passeriformes are not important in the current influenza emidemic, but are markedly affected by Usutu virus.

Due tot he high relevance of passeriformes for the epidemiology of West-Nile and Usutuviruses, the paragraph concering flaviviridae could be ammended by the data included in the previously mentioned reference, as well as the recent advancement of a subclustering system for West-Nile-Viruses: Santos (2023) An advanced sequence clustering and designation workflow reveals the enzootic maintenance of a dominant West Nile virus subclade in Germany. Virus Evol. 17;9(1):vead013. doi: 10.1093/ve/vead013

Concerning Retroviruses, the reported finding of diffuse lymphoblatic infiltration in liver, spleen and/or kidneys in five passeriformes by Watsworth et al. (1981), which would be diagnosed as lymphoma today is by no means a hint or proof of retroviral infection. In my view this is spontaneous and idiopathic lymphoma unless othervise proven. Therefore the sentence: „ALV neoplasms are recorded from captive passerines (Tangara schrankii and Garrulax albogular), and other avian orders, but the records are based on inspection and histology, and data is lacking on the aetiological agent causing gross lesions [286].“ needs o be rephrased.

It would be a great benefit for the reader if the authors could include a conclusions paragraph concerning the practical approach of the current knowedge for risk assesment when dealing with passeriformes. I would like to know, which biosafety countermeasures should be taken when ornithologists, clinicians and pathologists are handling with passeriormes? As example, we have in-house guidelines to wear FFPE masks and eye protection when dealing with wild doves and psittacids, because of the high prevalence of Chlamydia spp. Comparably, we started to wear increased personal protective equipment when dealing with highly West-Nile-Virus susceptible species like falcons, crows, owls and flamingos at least from June to October. What are your general recommendations concerning Passeriformes?

This manuscript is a review of 37 pages with 289 citations. I am not able to provide a thorough review including grammar and a cross check of this vast amount of literature in the short period of time given by this journal. Therefore, I just did a simple check of five randomly selected publications, and infer that the authors did their best with all the others and used a reference manager, preventing misreferencing. Furthermore, my grammar checker reports multiple minor alerts, mainly incorrect usage of singular and plural, as well as some duplications. Since I am not a native English speaker, the authors and the editors should use the grammar checker of their choice in order to correct these minor grammatical errors in the manuscript.

I hope that the authors will find at least some of my comments helpfull in order to improve and focus this manuscript, and hope that a thoroughly revised version will be published in one of the future issues of microorganisms.

This manuscript is a review of 37 pages with 289 citations. I am not able to provide a thorough review including grammar and a cross check of this vast amount of literature in the short period of time given by this journal. Therefore, I just did a simple check of five randomly selected publications, and infer that the authors did their best with all the others and used a reference manager, preventing misreferencing. Furthermore, my grammar checker reports multiple minor alerts, mainly incorrect usage of singular and plural, as well as some duplications. Since I am not a native English speaker, the authors and the editors should use the grammar checker of their choice in order to correct these minor grammatical errors in the manuscript.

Author Response

Dear Reviewer 1,

Thank you for a highly thought-provoking and thorough review, that helped us to improve the content of this review, as well as catching a few silly errors. 

Please find a detailed response to your comments below:

Reviewer 1: I feel that the authors overemphasize the potential danger of emerging and zoonotic diseases from passerine birds in this manuscript.

Response: Agreed. There are 3 instances of “zoonosis/zoonoses” and 18 of “zoonotic” (excluding references), but including, title and key words. We have reviewed each sentence. We feel that several viral taxa associated with passerines are zoonotic. It is important to state that some taxa are zoonotic, and that understanding potential zoonoses in wildlife is important. Some taxa are especially important as zoonosis: West Nile virus, Japanese encephalitis virus, etc. Where there is unequivocal evidence of zoonosis, we feel it is important information to supply to readers, and we have left that information as it is.

We include a new paragraph to explain the term zoonosis, and the relative importance of mammals versus birds as zoonotic reservoirs (114-119):

Greater than two-thirds of viral taxa that infect humans are considered to be zoonotic: they are able to infect non-human vertebrates, and may circulate in non-human reservoirs [18]. The alternative hosts for most zoonotic viruses are mammals (rodents, ungulates, other primates, carnivores, and bats). Birds are a much less important reservoir for zoonotic disease than mammals. Though less than 20% of zoonotic viruses share avian hosts [18], current data shows that birds are an important potential source for zoonotic viruses.

We also removed or altered the terms “zoonotic” (4 instances) or “emergent” (7 instances) where, on reflection, we thought it was weak (in the case of poorly studied and recently detected viruses) , or confused, as follows:

Line 37-38: we have removed “and almost all of them have zoonotic potential

Line 103-104: “Diverse pathogens, including emerging viruses, are probably transmitted between wild birds and domestic birds and poultry”. We removed “,including emerging viruses,”.

Line 133-134: we changed “which boosts the risk of the emergence of viral zoonotic diseases in humans [19] and poultry” change to “which boosts the risk of the circulation of viral diseases between humans [19], poultry and passerines

Line 142-144: It seems evident that the interface between agriculture and wildlife can be a trigger for the spread of new and emergent viruses among different species of birds, increasing the risk of pathogen spillover from domestic chickens, mainly backyard chickens, to wild birds [21] but probably vice versa as well.”  Has been changed to “It is possible that the contact zone between agriculture and wildlife provides an interface where viruses could potentially circulate between wild and domestic birds [23]”.

Line 401-402: Beak and feather disease virus (BFDV) is considered an emergent pathogen which can cause acute or subclinical disease in captive and wild psittacine species”. Change to “Beak and feather disease virus (BFDV) can cause acute or subclinical disease in captive and wild psittacine species”.

Line 589-591: Avian metapneumovirus (aMPV) is transmitted directly and causes upper respiratory tract diseases and reproductive disorders mainly in poultry, in which it is considered an emerging pathogen, and its epidemiology, dispersal patterns, and genetic evolution are not widely known [139]”: we have removed  “, in which it is considered an emerging pathogen,”.

Line 844-845: “MAYV was first isolated in 1954 in Trinidad [217]. It has caused outbreaks in several countries of the Caribbean and South America and is an emerging virus in the Americas [218].” We removed  and is an emerging virus in the Americas”.

Line 846-847: “There are no recent studies on potential non-human reservoirs, although need for increased surveillance for this emerging virus has been emphasized [224].” We removed for this emerging virus”.

Line 1055-1062:As well as posing a threat to poultry and a wide range of wild birds, rotaviruses may have zoonotic potential. RVA genotype G3, which is considered the third most prevalent genotype has a very broad host range, including artiodactyls, carnivores, chiropterans, leporids, perissodactyls, rodents, simians (including humans), and passerines [280]. Similarly, RVA genotypes G1 and G6 have been documented in a wide host range [279]. These findings suggest that interaction between humans and wild passerines, potentially due to environmental disturbances [281], or through illegal commercialization, can lead to spillover of rotaviruses to new hosts [282]”. Change to “As well as posing a threat to poultry and a wide range of wild birds, it has been suggested that rotaviruses may have zoonotic potential. RVA genotype G3, which is considered the third most prevalent genotype, has a very broad host range, comprising artiodactyls, carnivores, chiropterans, leporids, perissodactyls, rodents, simians (including humans), and passerines [280]. Similarly, RVA genotypes G1 and G6 have been documented in a wide host range [279]. Some authors have suggested that interaction between humans and wild passerines, potentially due to environmental disturbances [281], or through illegal commercialization, can lead to spillover of rotaviruses to new hosts [282]”.

Line 1063-1064: We removed “and strikingly, there is some suggestion that they may be zoonotic“ form “Due to their broad host range, and their negative consequences on poultry AvRVs are considered emerging pathogens [281] and strikingly, there is some suggestion that they may be zoonotic.

Reviewer 1 suggested that we improve our definition of One Health.

Response: We have added the following: Line 52-55 All these inter-dependent factors imply that the study of new or emerging viruses must be approached from a multisectoral and multidisciplinary perspective, framed in the One Health approach supported by the WHO, WOAH and FAO (4,5, Fig. 1).

Reviewer 1 felt in contrast, that there is reasonable concern of extinction of individual passeriform-species, and suggested that we discuss this subject, and suggested supporting literature. We added a short introduction to this theme

There is also reasonable concern that more vulnerable individual species (of all taxa, including Passeriformes) may be at risk of extinction from viral pathogens. It is suggested that island endemic species are particularly vulnerable to pathogens, especially introduced pathogens to which they have no prior contact and no innate im-munity [7]. Other authors have pointed out that all species with small geographic range, low population size and low genetic diversity may be highly vulnerable to ex-tinction, not just those island endemic species [8]. Theoretically species with small population size are vulnerable to disturbance, including pandemic disease. However, we are not aware of any evidence that any species has ever become extinct due to a viral pathogen.” (lines 58-66), citing one article already included in this review, and an article recommended by the reviewer (Manne, L., Brooks, T. & Pimm, S. Relative risk of extinction of passerine birds on continents and islands. Nature 399, 258–261 (1999). https://doi.org/10.1038/20436.

On the other hand, we did not feel that it was within the scope of this (very long) review to add a discussion on whether Passeriformes / birds in general are in danger of extinction, as suggested by the reviewer.

Reviewer 1: I suggest to include a clearly-stated and literature-backed definition of what is an „emerging disease“ in the introduction. I feel the authors underrate the „disease“, e.g. lesions, suffering, death, … in their current approach.

Response: With all due respect, we feel we clearly state “emergent disease” in the first paragraph of the introduction (lines 30-39), following 5 different references. Additionally, we discuss a number of complications to the definition, especially that powerful modern techniques mean that viruses are often detected, without the possibility of knowing if they cause disease or not. Our manuscript is about “emerging and novel viruses”, (the title of our manuscript)  which may cause “emerging and novel viral diseases”, but do not always. We have tried to tighten up our language (removing unnecessary or imprecise uses of zoonosis/emergent”.

Reviewer 1: For example, the sentence in the abstract: “ But adenoviruses, paramyxoviruses, coronaviruses, retroviruses, poxviruses, and herpesviruses could also be considered as emerging viruses.“ This is not appropriate and not backed by any available epidemiological data.

Response: We have changed this to “Arguably poxviruses, and perhaps other virus groups, could also be considered “emerging viruses”.(Lines 21-22)

We have emphasized the data showing that poxviruses are emerging diseases (using  references already cited, such as Paridae Poxvirus (Lawson, 2012): “infected with the emerging highly pathogenic Paridae Poxvirus [42,43].” (lines 272-273)

And “and “Paridae Poxvirus” in Europe” (line 311)

Reviewer 1: The introduction needs to be amended by an overview on the major families and genera of birds within the order passeriformes. This will markedly help to increase the understandability for the non ornithologist reader.

Response: That’s a very good suggestion. We have spent a considerable amount of time attempting to do this in a concise manner…difficult to do as we mention 99 passerine species of passerines from 30 families.

We have changed the first Passeriformes paragraph 121 – 130. Most of the changes are cosmetic….ordering of the sentence, or choice of words. However, we also introduce the question  “This begs the questions of whether the diversity of viruses circulating in Passeriformes is approximately equal to their share of avian diversity and abundance, and whether they could pose a significant risk to human and animal health and environmental balance” (lines 127-130).

We add two additional paragraphs, which describe the number of bird species reported in this review, and a brief introduction to the six families we discuss most frequently (lines 153-177). Additionally, we submit 2 new supplementary tables (Table S1: list of bird species used in this manuscript; Table S2: list of bord families used). These tables are intended to help the reader to understand the manuscript (and are not intended to be a comprehensive list of all viruses detected in all passerine species).

Reviewer 1: At the end of the 1st subsection, The authors state their hypothesis: „It seems evident that the interface between agriculture and wildlife can be a trigger for the spread of new and emergent viruses among different species of birds, increasing the risk of pathogen spillover from domestic chickens, mainly backyard chickens, to wild birds [21] but probably vice versa as well.  Passerines could play a larger role than previously thought in this interaction, a hypothesis that needs to be further studied.“ This sounds good, but a critical evaluation reveals that this is not the real topic of this review, and it can’t be the aim of a review. Which experimental methods and tests would be appropriate to aim at this interesting hypothesis? I guess, you would need paired samplings of poultry an passerine species surrounding the detection of a new and emerging virus – in a set previously known to be negative. I cannot find information like this somewhere in the manuscript.  I suggest to clearly state a hypothesis and aims for this manuscript, which follow the following guidelines: 1.) an aim needs to be specific, 2.) measurable as being reached or not in your final coclusions, 3.) interesting for the reader, 4.) achievable within this manuscript.

Response: Overall, we feel it will be difficult, if not impossible to find measurable data on the scale of this paper, and that it is thus not achievable within this manuscript. Furthermore, this is a review paper: we are presenting information on the viruses of passerines, and we do not feel that revisions should necessarily be hypothesis driven. However, we agree that our aims were not stated very clearly, and have revised them in the last paragraph of the introduction: “This review provides a detailed appraisal of the literature on viruses present in passerines, based on data from serosurvey, molecular surveillance, metagenomic studies and experimental inoculation studies. The data is ordered alphabetically following the unbiassed criteria of the type of viral genome and whether the virion is enveloped or naked. The authors also assess whether the available data demonstrates that the virus group should be considered novel or emerging. Since many of the findings are from metagenomic studies, an effort is made to differentiate those instances in which a link is established between disease and virus from those others which are incidental.” (lines 184-196).

Reviewer 1: Figure 1 is not balanced. Passerine species are presented like a parasite or health hazard. In my eyes, this is not appropriate or based on evidence. I suggest that the arrows should be at least double-headed, or even better, take the passerines out of the center: Passerines are not a hazard.

Response: We agree. We did not mean to imply that passerines are the centre of a one way process. Thanks for pointing out that our figure was confusing. We have revised the figure to emphasize that virus transmission may be a two way process, with bidirectional arrows that depict circulation cycles. We have changed the colour of two unidirectional, which are intended to depict consequences (and not transmission). An abstract ring for One Health implications has also been added. We do leave passerines at the centre of the figure (this is a review on passerines, after all). However, we take pains to be less focussed on passerines as the “cause of the problem”.

We have changed the legend to “Figure 1. The theoretical role of passerine birds in the circulation and potential spillover of viruses and other infectious agents to other passerines (blue semi-circular arrow around the bird in the centre) domestic, other peridomestic and other wild birds (blue bidirectional arrows) and their relation to potential zoonotic risk or threats to biodiversity(orange unidirectional arrows). The relation of passerines to theoretical consequences, such as zoonotic risk or threats to biodiversity, are depicted using orange unidirectional arrows. The One Health concept is represented by the light grey ring. (line 148-152).

Reviewer 1: The statement „Moreover, the suspicion that passerines may be infected with a specific virus is sometimes based only in the detection of antibodies during surveys of wild and domestic birds [23]“ should be rephrased, taking the in most situations retrospective view of serology into account.

Response: We have replaced this: “Moreover, occasionally there is a risk that results of serosurvey are over-interpreted as evidence that passerines may be infected with a specific virus [28], though seropositivity may only evidence a historic immune response to the virus”. Line 185-188

Reviewer 1: Table 1 includes highly valuable data. I should be amended by a column concerning the occurrence of disease or subclinical infection, not only death. The table should include the link to the references, especially concerning disease, death and zoonotic risk.  The table should include the link to the references, especially concerning disease, death and zoonotic risk.

Response: Very helpful suggestions. We had discussed including references and signs of disease, but omitted them due to size constraints. However, we follow your suggestion by adding a references column, a clinical signs of disease column, and by removing the zoonotic risk column in favour of a number code. We have placed the table at the end of the manuscript, as it is so big (pages 20-24)

We have improved the quality of the table.

Reviewer 1: I suggest to add, shape of virion, size of virion, envelope or not, configuration of genome, single or doublestranded, size of genome, RNA or DNA for every family.

Response: Another very helpful suggestion. We have added this information for each virus family, at the start of the section.

Reviewer 1: Concerning Circoviridae and related viruses, the paragraph should be amended by one or two more sentences introducing what a CRESS is to the non-expert. Furthermore, follow the general guideline of first spelling out and adding an abbreviation afterwards in brackets, and not vice versa.

Response: Of course. We have added “Circular Rep-encoding single stranded (CRESS)” (line 424)

Reviewer 1: 144 HN combinations: change Line 417 and Line 419.

Response: Thanks for spotting that typo. Changed line 469 and 477.

Reviewer 1: the references concerning passeriformes as hosts of influenza viruses as well as important flaviviruses are rather old. I suggest to include some references targeting the recent epidemic in Europe: Günter et al. 2023. Emerg Microbes Infect. 12(2):2231561. This article highlights, that passeriformes are not important in the current influenza epidemic, but are markedly affected by Usutu virus.

Response: We have incorporated that very nice article, and modified the sections on AI and USUV accordingly.  “A recent study from the Germany, including data from 2006-2021, including data from the 2020-2021 highly pathogenic AIV outbreak found only 11/972 (1.1%) passerines (all Corvidae) to be positive, compared 3351/4583 (73%) of all birds tested [109]” (Line 491-493).

And “The above-mentioned avian virus study from Germany, including data from 2006-2021, found USUV in only 917/972 (94.3%) passerines to be positive, compared 1042/4583 (22.7%) of all birds tested [109]. The vast majority (82.2%) of positive birds were common blackbird..“ (lines 735-738) Information in the supplementary material of this article, furnished about 25 additional species for the USUV supplementary table (now supplementary Table S4). We have altered the text in the manuscript and incorporated the additional species into the table. “USUV has been detected in at least 100 bird species (Table S4), 46 species (46%) and 17 families from Passeriformes, somewhat lower than the proportion of bird species within the order.” (lines 739-740)

Reviewer 1: the paragraph concering flaviviridae could be ammended by the data included in the previously mentioned reference, as well as the recent advancement of a subclustering system for West-Nile-Viruses: Santos (2023) An advanced sequence clustering and designation workflow reveals the enzootic maintenance of a dominant West Nile virus subclade in Germany. Virus Evol. 17;9(1):vead013. doi: 10.1093/ve/vead013

Response: That’s a really interesting, detailed article. However, we feel that the WNV sub-clusters circulating in Germany in 2020 is beyond the scope of our manuscript.

Reviewer 1: Retroviruses: In my view this is spontaneous and idiopathic lymphoma unless othervise proven. The sentence: „ALV neoplasms are recorded from captive passerines (Tangara schrankii and Garrulax albogular), and other avian orders, but the records are based on inspection and histology, and data is lacking on the aetiological agent causing gross lesions [286].“ needs to be rephrased.

Response: This has been changed to “Lymphoid leukosis was recorded from captive passerines (Tangara schrankii and  Pterorhinus albogularis) based on gross post-mortem and histopathology findings [286], but the aetiological agent was not confirmed as ALV.” (Lines 1079-1081)

Reviewer 1: A conclusions paragraph concerning the practical approach of the current knowedge for risk assesment when dealing with passeriformes. I would like to know, which biosafety countermeasures should be taken when ornithologists, clinicians and pathologists are handling with passeriformes? 

Response: We have added Section 3 “General recommendations to reduce the risk of transmission of viruses associated with passerines” (lines 1097-1123) to give general advice on safe handling of passerines.

Reviewer 1: There are a lot of citations, and a quick check suggests that 1), they have been inserted using a reference manager; and 2) errors have not been found.

Response: We have tried hard to reduce errors in the references, but as you point pout there are roughly 290 citations. We continue to check and recheck the references, and hope we have eliminated all errors.

Reviewer 1: The grammar checker reports multiple minor alerts, mainly incorrect usage of singular and plural, as well as some duplications.

Response: I personally ran my grammar checker before submission – and it gave me the all clear! We have a found a few minor issues during the revision process. We have repeated grammar check: and it currently tells us there are no issues.

Reviewer 2 Report

The authors have written an extensive review of the infection of Passerine birds by different virus families. The review is extensive and thorough, describing what is currently known and more importantly, revealing the gaps in knowledge.

One suggestion for improvement is to make Table 1 more legible. The fonts used make it hard to read the table. Reformatting the table to improve clarity and legibility will be very helpful to the reader.

The legend to Figure 2 contains several virus abbreviations lacking an explanation (e.g. BFDV and CacV).

Author Response

Dear Reviewer 2,

Thank you for very positive review. Specifically you pointed out some silly ommissions, and the poor quality of the table, and we greatly appreciate your help.

We have made much more substantive changes than you suggested, on the recommendation of Reviewer 1. We authors feel that all these changes have greatly improved our manuscript.

Please find a detailed response to your comments below:

Reviewer 2: The authors have written an extensive review of the infection of Passerine birds by different virus families. The review is extensive and thorough, describing what is currently known and more importantly, revealing the gaps in knowledge.

One suggestion for improvement is to make Table 1 more legible. The fonts used make it hard to read the table. Reformatting the table to improve clarity and legibility will be very helpful to the reader.

Response: Thanks for pointing out this issue. We agree that the presentation of out table was not good enough. We have reformatted the table to improve clarity and legibility. Additionally, we follow the suggestion of Reviewer 1 by adding a references column, a clinical signs of disease column, and by removing the zoonotic risk column in favour of a number code. We have placed the table at the end of the manuscript, as it is so big (pages 20-24)

We have improved the quality of the table.

Reviewer 2: The legend to Figure 2 contains several virus abbreviations lacking an explanation (e.g. BFDV and CacV).

Response: We have reviewed the figure, and inserted the abbreviations that we had left out in the previous version (lines 255-264). This was also a problem for Figure 4, and we have taken the same actions (line 1161).

Reviewer 3 Report

This thorough review gathers information from peer-reviewed papers that describe detection of viruses in passerine birds. The authors are right, most research papers on passerine viruses are aimed at avian influenza and some flaviviruses while emerging viruses are overlooked, which is unfortunate and potentially threatening given the amount of illegal between-country bird trades that occur worldwide. For this reason, I appreciate the authors’ comprehensive work presented here.

The summary and intro lead the reader into thinking this is a literature review focusing in virome studies detected by metagenomics, but this paper is far more complete than that. As far as I understand, the use of the term “virome” implies shotgun sequencing of viral genomes that may or may not have clinical significance. In this study, the authors go further, including and emphasizing the importance of clinical relevance in association with lesions and clinical signs and barely mentions work using metagenomics. In other words, I don’t think this is a virome review, this is a paper about viral pathogens of passerines (which to me is more relevant than a virome review). Perhaps the summary, intro, and conclusions could be altered to reflect that.

The conclusion is very well thought and brings up sampling biases to address why some species seem to be more often tested (and consequently affected) than others. It provides a global overview on disease surveillance in passerine birds that I honestly haven’t thought about before reading this manuscript. In summary, this paper not only brings a well-put literature review on viral disease surveillance in passerines, but also provides the authors’ perspective about risks and interpretation of surveillance data.

My only suggestion is to present the virus families in the order of their relevance to passerine birds instead of grouping subsections based on virus genomic features. This is a long paper in which passerines are the most interesting feature, and not the virus families and classification.

Some very minor comments are listed below:

(1) The authors should make sure they have line numbering on the document when they submit a paper for peer review.

(2) Figure 1 is easy to understand because the paragraph that refers to it does a good job putting it in context. However, the blue arrows are confusing. This image could be improved to be more self-explanatory.

(3) Table 1 is really poor quality. It needs to be formatted in the Word document instead of added as an image.

(4) There is some conflicting information regarding the clades of canarypox and fowlpox viruses (3rd paragraph of section 2.1.3.).

(5) The subsection numbers go from 2.1.1. to 2.1.3.

(6) The page numbering is incorrect from page 6 on.

Author Response

Dear Reviewer 3,

Thank you for very positive review.  We greatly appreciated your very positive comments. You pointed out several little omissions or ways to improve our manuscript.

We have made much more substantive changes than you suggested, on the recommendation of Reviewer 1. We authors feel that all these changes have greatly improved our manuscript.

Please find a detailed response to your comments below:

Reviewer 3: The summary and intro suggest this literature review is focused on virome studies detected by metagenomics: but this is really a paper about viral pathogens of passerines (which to me is more relevant than a virome review). Perhaps the summary, intro, and conclusions could be altered to reflect that.

Response: We didn’t mean to give the impression that this is a virome review, this is a review on the viruses in passerines. We derive our data from many different types of studies: metagenomics, serosurvey, molecular surveillance, inoculation studies, etc.  The concept of “viroma” is central to our knowledge of some virus families (e.g.: Circoviridae, Adenoviridae or Astroviridae). Many of the papers used for the review are metagenomic studies of cloacal viroma in which the pathogenicity of many of these viruses has not been determined (e.g., 7, 13, 15, 258, 268, 269, 273, 276...). We agree that it will help the reader to be clearer.

We have added “from traditional and metagenomic studies” to the abstract (line 18).

We have emphasized in the final paragraph of the introduction we derive our data from beyond metagenomic virome studies: “based on data from serosurvey, molecular surveillance, metagenomic studies and experimental inoculation studies” (lines 190-191).

Reviewer 3:  My only suggestion is to present the virus families in the order of their relevance to passerine birds instead of grouping subsections based on virus genomic features. This is a long paper in which passerines are the most interesting feature, and not the virus families and classification.

Response: An interesting suggestion, though we disagree. Too little is known about the relevance of virus families to passerine birds so the order would be easy to establish only in the case of the most relevant and best studied avian families. This would ultimately become the relevance of passerine viruses to humans. We feel it is easiest for the reader to find the virus families, which follow an internationally accepted order supported by the ICTV and the Baltimore classification.

Reviewer 3: The authors should make sure they have line numbering on the document when they submit a paper for peer review.

Response: Corrected

Reviewer 3: Figure 1 is easy to understand because the paragraph that refers to it does a good job putting it in context. However, the blue arrows are confusing. This image could be improved to be more self-explanatory

Response: Thanks for pointing out that the blue arrows in Figure 1 are confusing. We have revised the figure to emphasize that virus transmission may be a two way process, with bidirectional arrows that depict circulation cycles. We have changed the colour of two unidirectional, which are intended to depict consequences (and not transmission). An abstract ring for One Health implications has also been added. We leave passerines at the centre of the figure.

We have changed the legend to “Figure 1. The theoretical role of passerine birds in the circulation and potential spillover of viruses and other infectious agents to other passerines (blue semi-circular arrow around the bird in the centre) domestic, other peridomestic and other wild birds (blue bidirectional arrows) and their relation to potential zoonotic risk or threats to biodiversity(orange unidirectional arrows). The relation of passerines to theoretical consequences, such as zoonotic risk or threats to biodiversity, are depicted using orange unidirectional arrows. The One Health concept is represented by the light grey ring. (line 148-152).

Reviewer 3: Table 1 is really poor quality. It needs to be formatted in the Word document instead of added as an image.

Response: It was very poor quality. We have fixed this. Additionally, we follow the suggestion of Reviewer 1 by adding a references column, a clinical signs of disease column, and by removing the zoonotic risk column in favour of a number code. We have placed the table at the end of the manuscript, as it is so big (pages 20-24)

Reviewer 3: There is some conflicting information regarding the clades of canarypox and fowlpox viruses (3rd paragraph of section 2.1.3.).

Response: Well spotted. Fixed. It now reads “Clade A (Fowlpox virus, FWPV), Clade B (represented by the species Canarypox virus, CNPV)” (lines 297-298)

Reviewer 3: The subsection numbers go from 2.1.1. to 2.1.3.

Response: Fixed

Reviewer 3: The page numbering is incorrect from page 6 on.

Response: Fixed